

# EFT asymptotics: The growth of operator degeneracy

**Tom Melia[1]$^\star$ and Sridip Pal[2]$^\dagger$**

**1** Kavli Institute for the Physics and Mathematics of the Universe (WPI),
UTIAS, The University of Tokyo, Kashiwa, Chiba 277-8583, Japan
**2** School of Natural Sciences, Institute for Advanced Study, Princeton, NJ 08540, U.S.A.

$\star$ tom.melia@ipmu.jp, $\dagger$ sridip@ias.edu

## Abstract

We establish formulae for the asymptotic growth (with respect to the scaling dimension) of the number of operators in effective field theory, or equivalently the number of $S$-matrix elements, in arbitrary spacetime dimensions and with generic field content. This we achieve by generalising a theorem due to Meinardus and applying it to Hilbert series—partition functions for the degeneracy of (subsets of) operators. Although our formulae are asymptotic, numerical experiments reveal remarkable agreement with exact results at very low orders in the EFT expansion, including for complicated phenomenological theories such as the standard model EFT. Our methods also reveal phase transition-like behaviour in Hilbert series. We discuss prospects for tightening the bounds and providing rigorous errors to the growth of operator degeneracy, and of extending the analytic study and utility of Hilbert series to EFT.



# 1 Introduction

The principle of effective field theory (EFT) is to parameterise all possible contributions to the $S$-matrix that can occur in a quantum field theory of given particle content and symmetry. Its phenomenological utility relies on a hierarchy of experimental importance of these contributions, and on the possibility of probing higher order terms via an improvement in experimental precision. Both of these aspects are necessary, for example, for the search for new physics at the Large Hadron Collider within the framework of the standard model EFT (SMEFT) to be a viable endeavour. The rate of growth of the number of independent $S$-matrix contributions that can be constrained with increasing experimental precision is an important practical issue: even at the lowest few orders in the SMEFT expansion, their number is cumbersome [1–3]. Experimental practicalities aside, the results of this paper can be used to determine that at around the 120th order in the SMEFT expansion, the number of possible independent measurements that can be performed exceeds the number of atoms in the observable universe (around $10^{80}$). Our purpose, however, is not to establish where such cosmologically large thresholds lie. Rather it is to develop new analytic methods by which such numbers can be obtained, and to explore what these techniques can reveal about EFT itself through its analytic study.

The analytic study of EFTs/$S$-matrices dates back to the discovery of the strong interaction and the birth of $S$-matrix theory (*e.g.* [4]). Many of the ideas and methods have found utility in modern amplitude calculations in perturbative QCD and other theories; for reviews see

*e.g.* [5, 6]. In the past few years, building upon the success of the modern CFT bootstrap program [7], a number of new analytic results have been obtained in an *S*-matrix bootstrap approach [8–19]. Here, we focus on a much simpler mathematical object that has recently been introduced to study EFT/*S*-matrices—Hilbert series.

Hilbert series have appeared in the particle physics literature as partition functions to enumerate gauge and flavour invariants [20–24], and were applied to EFT operator degeneracy counting in [3, 25–30]. A study of their utility as analytic probes was initiated in [26, 28], establishing, for example, recursion relations between Hilbert series for theories of different field content in $d = 2$ spacetime dimensions, and all-order in derivatives results for degeneracy of *S*-matrix elements with a low fixed number of external particles. The aim of this paper is to extend what we know analytically about Hilbert series, and thus EFTs/*S*-matrices, by extracting the full asymptotic growth (with respect to the scaling dimension) of number of operators from the Hilbert series, going beyond the basic studies on asymptotic growth presented in [28]. From now on, by "asymptotic growth of operators" or "asymptotic operator growth" , we always mean the growth of number of operators with scaling dimension $\Delta$ as $\Delta \to \infty$.

Appropriately graded Hilbert series correspond to partition functions of free quantum field theories. There is thus a connection to many of the techniques to study partition functions that appear prominently in the study of CFT. The methods and formulae established by Cardy [31] provide the growth of operator degeneracy in $d = 2$. We review the use of modular invariance of the partition function for $d = 2$ scalar EFT to obtain the asymptotic growth of all operators, and those captured by the Hilbert series (i.e. a projection from the space of all operators to the spin zero subspace). In $d = 2$, we can also utilize the connection of the Hilbert series to integer partitions, as established in [26], to capture asymptotic growth via the famous Hardy-Ramanujan formula.

In higher dimensions and including more general field content—fermionic and higher spin representations, possibly with internal symmetry degrees of freedom—modular invariance of the partition function is lost, and much less is known. Leading behaviour of the free field partition function (for all operators) of a scalar field in arbitrary $d$ was presented in [32]. We develop and apply a theorem due to Meinardus, so as to obtain exact results for the asymptotic growth of partition functions of the more general EFTs mentioned above. We apply our results to obtain the asymptotic growth of operators in the SMEFT. Somewhat surprisingly, the asymptotic formulae we obtain are in remarkably good agreement with exact results for operator degeneracy at low scaling dimension.

In line with our above goal of analytic exploration, we also explore subtleties that arise in evaluating the saddle points when projecting onto a singlet sector—phase transition-like behaviour is observed in Hilbert series, e.g. in taking the limit where the number of fermions goes to zero.

The analytic techniques and results developed here could further have application to the study of large particle number and high temperature limits of gases of non-interacting particles, *e.g.* in determining level densities [33], and multiplicity distributions in models of hadrons in very high-energy particle collisions [34, 35].

As a way of defining the techniques we use to mine asymptotics and introducing Hilbert series themselves, consider a quantum field theory with a discrete spectrum $\{\Delta_n\}$ and corresponding degeneracies $\{a_n\}$, which means that there are $a_n$ number of operators with scaling dimension $\Delta_n$. The Plethystic Exponential (PE) is defined as the sum,

$$PE(q) = \sum_n a_n q^{\Delta_n} . \tag{1}$$

The $PE(q)$ can also be though of as generating function of number of operators. We will work

with the variables $q = e^{-\beta}$. To estimate the growth of the degeneracies, we write

$$PE(q) = \sum_n a_n e^{-\beta \Delta_n} \equiv \int_0^\infty d\Delta \, \rho(\Delta) e^{-\beta \Delta} \,, \tag{2}$$

where

$$\rho(\Delta) \equiv \sum_n a_n \, \delta(\Delta - \Delta_n) \,. \tag{3}$$

Thus formally we have

$$\rho(\Delta) = \frac{1}{2\pi\iota} \int_{\Gamma - \iota\infty}^{\Gamma + \iota\infty} d\beta \, PE\big(q = e^{-\beta}\big) e^{\beta \Delta} \,. \tag{4}$$

Our objective is to estimate asymptotic growth of $a_n$ for large $n$ (or equivalently $\rho(\Delta)$ for large $\Delta$). The idea is that the asymptotic growth of $a_n$ is encoded in the behaviour of PE in the $q \to 1$ limit[1] [32]. In particular, we have [31,32]

$$\rho(\Delta) \underset{\Delta \to \infty}{\simeq} \frac{1}{2\pi\iota} \int_{\Gamma - \iota\infty}^{\Gamma + \iota\infty} d\beta \, PE(q \to 1) e^{\beta \Delta} \,. \tag{5}$$

For EFT we typically want to count operators which are singlets under the Lorentz symmetry group and possibly some internal symmetry group. To do this, one needs to turn on variables (fugacities), $w_i$, for these groups and project out only the singlet terms from the PE integrating over the group measure, schematically

$$H(q) = \int d\mu_{\text{Lorentz}} \int d\mu_{\text{internal}} \frac{1}{P(q, w_i)} PE(q, w_i) \,, \tag{6}$$

where $d\mu$ are the Haar measures of the symmetry groups, and where we also included a projector $P(q, w_i)^{-1}$ (the inverse of the momentum generating function) to count only classes of operators equivalent up to a total derivative, see [28] for further details. Operators within each class are said to be related by integration by parts (IBP); there is an equivalence class of IBP related operators for each conformal primary operator in the spectrum. Imposing IBP in a Hilbert series means throwing out the descendant operators from the counting [28].

The above produces a Hilbert series $H(q)$, which is a function of $q = e^{-\beta}$. We are interested in obtaining asymptotic form of this generating function in the $\beta \to 0$ limit, and in what follows we will show it takes the form

$$H(\beta \to 0) = \exp\left[ \sum_{k=0}^{d-1} a_k \beta^{-k} + b \log(\beta) + c \right], \, a_{d-1} > 0 \,,$$

where $d$ is the space-time dimension. Again one can write,

$$H(\beta) = \int_0^\infty d\Delta \, \rho_g(\Delta) e^{-\beta \Delta} \,, \tag{7}$$

where $\rho_g$ is the density of number of operators in the singlet sector. Similar considerations to the above yield

$$\rho_g(\Delta) \underset{\Delta \to \infty}{\simeq} \frac{1}{2\pi\iota} \int_{\Gamma - \iota\infty}^{\Gamma + \iota\infty} d\beta \, H(\beta \to 0) e^{\beta \Delta} \,. \tag{8}$$

---

[1]The intuition behind this is that $q \to 1$ limit, $PE(q)$ diverges. Formally $PE(q = 1) = \sum_n a_n$ diverges too. Thus the $PE(q \to 1)$ encodes the growth of $\sum_{n=1}^N a_n$ as $N \to \infty$, hence the growth of $a_n$ as $n \to \infty$.

The inverse Laplace transformation can be performed using saddle point approximation. For large enough $\Delta$, the saddle will be $\Delta_* \propto \beta^{-d}$.

A careful reader will notice that while the right hand side of Eq. (5) and Eq. (8) is a continuous function, the left hand side is a distribution. The proper way to interpret this is to smear $\rho(\Delta)$ over a small window of $[\Delta - \delta, \Delta + \delta]$ and then estimate the number of states lying in that window as $\Delta \to \infty$, keeping $\delta$ fixed. The mathematical machinery that one needs to achieve this lies in the Tauberian theorems (See [36] and appendix C of [37] for a basic introduction) In 2D conformal field theories (free and/or interacting) it is possible to estimate the density of states using these techniques and even put an upper bound on the spectral gap in interacting CFTs [38–41]. We will not attempt to make our general analysis mathematically rigorous in the sense just described and leave it as a future avenue to explore.

The remainder of this paper is organised as follows. In Sec. 2 we review how to obtain rigorous results on the growth of operators in $d = 2$ using modular properties of the PE, and detail the connection to the Hardy-Ramanujan formula. Moving away from $d = 2$ the modular properties are lost. In Sec. 3 we introduce a new trick that can quickly obtain asymptotic formulae for the growth of operators for more general partition functions, up to an unknown order one number; we test its accuracy by comparing to the known asymptotic growth of plane partitions, and we show how the PE for a real scalar field in $d = 3$ can be related to the partition function of plane partitions. In Sec. 4 we obtain our main results: by generalising a theorem of Meinardus, we find exact asymptotic formulae that can be applied to the PE for EFTs in general $d$ and including particles of spin. Sec. 5 presents the saddle point techniques used to make the projections of the PE to singlets of spacetime and internal symmetries, and we provide general formulae for the resulting Hilbert series. In Sec. 6 we detail a subtlety in taking the saddle point approximation when fermions are present, and observe phase transition-like behaviour in the Hilbert series. We apply our results to the SMEFT in Sec. 7, and observe remarkable numerical agreement at low mass dimension in this theory. Sec. 8 contains a discussion of refinements and generalisations that can be made to the analysis we present here.

We highlight some of our main results: Eq. (102) for the leading behaviour of the free field partition function in arbitrary $d$ and for arbitrary spin $j$; the general lessons in 5.2, in particular, Eq. (116) for the asymptotic growth of operators in EFTs in four dimensions, including the exact order one multiplicative factors; and, Eq. (145) for the asymptotic growth of operators in the SMEFT.

## 2 Rigour in two dimensions

We begin by considering a single scalar field $\phi$ in $1 + 1$ dimensions. This section follows the analysis and methods introduced in [31]. We will first estimate the growth of all the operators as encoded in the Plethystic exponential. Using an appropriate projector, the PE can be turned into a Hilbert series, which encodes the number of scalars appearing in the theory; we will move on to estimating the growth of scalars using this Hilbert series. Finally, we will impose the IBP constraints, which amounts to throwing out the $SL(2, R)$ descendants; again our aim is to be find an estimate of the growth of operators.

The Plethystic exponential has the information of all the operators that one can construct, not necessarily invariant under Lorentz group. It is given by

$$PE = \exp\left[\sum_{n=1}^{\infty} \frac{1}{n}\left(\chi_\phi(t^n, x^n) - 1\right)\right] = \exp\left[\sum_{n=1}^{\infty} \frac{1}{n}\left(\frac{(1 - t^{2n})}{[1 - (tx)^n][1 - (t/x)^n]} - 1\right)\right], \quad (9)$$

where $\chi_\phi(t, x)$ is the character for the spin-0 conformal representation, see e.g. [42], with

fugacities $t$ for scaling dimension (which effectively counts derivatives) and $x$ for angular momentum.[2]

We emphasise the presence of the $-1$ in the above equation is needed to define the Plethystic exponential for the case $d = 2$. Without the $-1$, the PE is not well-defined *i.e.* it has a divergence when setting $t = 0$ of the form $\exp\left[\sum_n \frac{1}{n}\right]$. From a physics point of view, this divergence has a meaning. Its origin lies in the fact that in $d = 2$, $\phi$ is dimensionless and one can construct an arbitrarily large number of operators using powers of $\phi$ without changing the scaling dimension. To be precise, given an operator $\mathcal{O}$, one can construct arbitrary number of operators by considering $f(\phi)\mathcal{O}$, without changing the scaling dimension. We form an equivalence class by counting two operators only once if they differ only by their number of $\phi$s, and thus $f(\phi)\mathcal{O}$ and $\mathcal{O}$ are not counted separately. The implementation of this equivalence relation boils down to defining the PE by throwing away the term $\exp\left[\sum_{n=1}^{\infty} \frac{1}{n}\right]$, hence introducing a $-1$ in the definition of $PE$. In higher dimension, we do not need the factor of $-1$ to define the $PE$; this feature is unique to $d = 2$ where $\phi$ is dimensionless.

We will be using the variables $\beta$ and $\omega$,

$$t = e^{-\beta}, \ x = e^{2\pi\iota\omega}, \tag{10}$$

and for use in the below we define the variables $q$ and $\bar{q}$,

$$q = e^{-\beta - 2\pi\iota\omega}, \ \bar{q} = e^{-\beta + 2\pi\iota\omega}. \tag{11}$$

We rewrite the PE in the following fashion

$$PE = \exp\left[\sum_{n=1}^{\infty} \frac{1}{n}\left(\frac{1}{1-q^n} - 1\right)\right]\exp\left[\sum_{n=1}^{\infty} \frac{1}{n}\left(\frac{1}{1-\bar{q}^n} - 1\right)\right]. \tag{12}$$

## 2.1 Analysis via modular properties of the Dedekind eta function

The regularized PE can be related to Dedekind eta function as

$$PE = \frac{q^{1/24}\bar{q}^{1/24}}{\eta(q)\eta(\bar{q})}. \tag{13}$$

We remark that the regularized PE is related to the torus partition function of $c = 1$ free bosonic CFT modulo the zero mode contribution (see e.g. Chapter 10 of [43]).

**All the operators**

To study all the operators, we can turn the angular momentum fugacity off, setting $q = \bar{q}$. In order to extract the asymptotics, we must know $\eta(q)$ in the $q \to 1$ limit. The nice feature is that the $\eta$ function has a modular property

$$\eta(q = e^{-\beta}) = \sqrt{\frac{2\pi}{\beta}}\eta\left(\tilde{q} = e^{-\frac{4\pi^2}{\beta}}\right), \tag{14}$$

and

$$\eta(\tilde{q} \to 0) = \exp\left[-\frac{\pi^2}{6\beta}\right]. \tag{15}$$

---

[2]The presence of the $(1 - t^2)$ in the conformal character enacts the null state condition—this is the condition of imposing equations of motion (EOM) on operators in an EFT (see [28] for a detailed discussion). All the fields of various spin that we will consider contain such null states (they saturate a unitarity bound in the free theory); we present a short discussion on the effect that EOM have on operator growth in App. A.

Thus in $\beta \to 0$ i.e. $q \to 1$ limit, we have

$$\lim_{q \to 1} PE = \lim_{q \to 1} \eta(q)^{-2} = \frac{\beta}{2\pi} \lim_{\tilde{q} \to 0} \eta(\tilde{q})^{-2} = \frac{\beta}{2\pi} \exp\left[\frac{\pi^2}{3\beta}\right]. \tag{16}$$

Taking the inverse Laplace transform of the above, as per Eq. (5), we obtain the asymptotic growth of all the operators, $\rho(\Delta)$, as

$$\rho(\Delta) \underset{\Delta \to \infty}{\simeq} \text{Inverse Laplace}\left[\lim_{q \to 1} PE\right] = \frac{\pi^3}{18} \, {}_0\tilde{F}_1\left(3; \frac{\pi^2 \Delta}{3}\right) \simeq \frac{3^{-3/4}}{4} \frac{e^{\frac{2\pi\sqrt{\Delta}}{\sqrt{3}}}}{\Delta^{5/4}}. \tag{17}$$

## Scalar operators

The analysis for counting the scalar operators is done by projecting the $PE(q, \bar{q})$ onto spin 0. This is done by contour integral of the PE over the maximal compact subgroup of $SL(2, R)$, namely $U(1)$, with Haar measure

$$\oint d\mu_{U(1)} = \oint_{|x|=1} \frac{dx}{2\pi \iota x}. \tag{18}$$

Here we will work with the real variable $\omega$ such that

$$H = \int_{-1/2}^{1/2} d\omega \, PE(q, \bar{q}). \tag{19}$$

The asymptotic analysis of $H$ can be done following the fixed spin analysis of [41]. In this case, we will not set $q = \bar{q}$, instead keeping them complex conjugate to each other. Now we have

$$PE(q, \bar{q}) = \frac{e^{-\frac{\beta}{12}}}{\eta(q)\eta(\bar{q})} = \sqrt{\frac{\beta + 2\pi\iota\omega}{2\pi}} \sqrt{\frac{\beta - 2\pi\iota\omega}{2\pi}} \frac{e^{-\frac{\beta}{12}}}{\eta(\tilde{q})\eta(\bar{\tilde{q}})}$$

$$\underset{\beta \to 0}{\simeq} \frac{\sqrt{\beta^2 + 4\pi^2\omega^2}}{2\pi} e^{\frac{4\pi^2}{\beta + 2\pi\iota\omega} \frac{1}{24} + \frac{4\pi^2}{\beta - 2\pi\iota\omega} \frac{1}{24}} = \frac{\sqrt{\beta^2 + 4\pi^2\omega^2}}{2\pi} e^{\frac{\pi^2 \beta}{3(\beta^2 + 4\pi^2\omega^2)}}. \tag{20}$$

Thus we have

$$H(\beta) \underset{\beta \to 0}{\simeq} \int_{-1/2}^{1/2} d\omega \, \frac{\sqrt{\beta^2 + 4\pi^2\omega^2}}{2\pi} e^{\frac{\pi^2 \beta}{3(\beta^2 + 4\pi^2\omega^2)}}. \tag{21}$$

Now note that in $\beta \to 0$ limit, the integral is dominated by $\omega = 0$ saddle. Thus we can do a saddle point approximation around $\omega = 0$ and obtain

$$H(\beta) \underset{\beta \to 0}{\simeq} \frac{\beta}{2\pi} e^{\frac{\pi^2}{3\beta}} \int_{-\infty}^{\infty} d\omega \, \exp\left(-\frac{4\pi^4\omega^2}{3\beta^3}\right) = \frac{\sqrt{3}\beta^{5/2}}{4\pi^{5/2}} e^{\frac{\pi^2}{3\beta}}. \tag{22}$$

Thus we can see that while the leading exponential growth stays the same, the projection onto scalar operators changes the polynomial dependence on $\beta$. There is an extra factor of $\sqrt{\frac{3\beta^3}{4\pi^3}}$. The growth of scalar operators is be suppressed by $3^{-1/4}\Delta^{-3/4}$ compared to the growth of all the operators. This comes from noting that the inverse Laplace transformation is dominated by the saddle $\beta = \pi\sqrt{\frac{c}{3\Delta}}$. We can also verify the scaling by doing the inverse Laplace transformation of Eq. (22) explicitly:

$$\rho^{\text{scalar}}(\Delta) \underset{\Delta \to \infty}{\simeq} \frac{e^{2\pi\sqrt{\frac{\Delta}{3}}}}{12\Delta^2} = \left(\frac{1}{3\Delta^3}\right)^{1/4} \left(\frac{3^{-3/4} \exp\left(2\pi\sqrt{\frac{\Delta}{3}}\right)}{4\Delta^{5/4}}\right). \tag{23}$$

**IBP constraints**

To impose the IBP constraint, we include a projector that is the inverse of the momentum generating function

$$P(q,\bar{q})^{-1} = (1-q)(1-\bar{q}). \tag{24}$$

The regularized PE becomes

$$PE_{\text{IBP}} = (1-q)(1-\bar{q})\exp\left[\sum_{n=1}^{\infty}\frac{1}{n}\frac{q^n}{1-q^n}\right]\exp\left[\sum_{n=1}^{\infty}\frac{1}{n}\frac{\bar{q}^n}{1-\bar{q}^n}\right]. \tag{25}$$

First we look at the asymptotic growth of all operators, setting $q = \bar{q}$; the $\beta \to 0$ limit of $PE_{\text{IBP}}$ is given by

$$PE_{\text{IBP}} \underset{\beta \to 0}{\simeq} \frac{\beta^3}{2\pi}\exp\left[\frac{\pi^2}{3\beta}\right]. \tag{26}$$

Thus with the IBP constraint the growth of all operators are given by

$$\rho_{\text{IBP}}(\Delta) \underset{\Delta \to \infty}{\simeq} \frac{\pi^3 I_4\left(\frac{2\pi\sqrt{\Delta}}{\sqrt{3}}\right)}{18\Delta^2} \simeq \frac{\pi^2}{12}\frac{3^{-3/4}e^{\frac{2\pi\sqrt{\Delta}}{\sqrt{3}}}}{\Delta^{9/4}} = \left(\frac{\pi^2}{3\Delta}\right)\left(\frac{3^{-3/4}e^{\frac{2\pi\sqrt{\Delta}}{\sqrt{3}}}}{4\Delta^{5/4}}\right). \tag{27}$$

Compared to Eq. (17), we have suppression by $\beta$ i.e $\pi^2/3\Delta$. We will see that in $d$ dimensions, the role of IBP is to introduce an extra factor of $\beta^d$ in the $\beta \to 0$ limit of the PE of Hilbert series. This results in a generic suppression of $\rho(\Delta)$ by a factor of $\Delta^{-1}$ (since the saddle point of the inverse Laplace transform is given by $\beta_*^d \propto \Delta^{-1}$) when comparing the growth of operators with IBP imposed to the growth of all operators.

**Scalars and IBP**

The introduction of IBP amounts to modification of Eq. (22) into following:

$$H_{\text{IBP}}^{\text{scalar}}(\beta) \simeq \frac{\beta}{2\pi}(1-e^{-\beta})(1-e^{-\beta})e^{\frac{\pi^2}{3\beta}}\int_{-\infty}^{\infty}d\omega\exp\left(-\frac{4\pi^4\omega^2}{3\beta^3}\right) \underset{\beta \to 0}{\simeq} \frac{\sqrt{3}e^{\frac{\pi^2}{3\beta}}\beta^{9/2}}{4\pi^{5/2}}, \tag{28}$$

where we have pulled out the factor accounting for the IBP constraint out of the $\omega$ integral evaluated at the saddle point $\omega = 0$. The asymptotic growth of scalar operators with IBP constrained imposed is then given by

$$\rho_{\text{IBP}}^{\text{scalar}}(\Delta) \underset{\Delta \to \infty}{\simeq} \frac{\pi^2}{36}\frac{e^{\frac{2\pi\sqrt{\Delta}}{\sqrt{3}}}}{\Delta^3} = \left(\frac{1}{3\Delta^3}\right)^{1/4}\left(\frac{\pi^2}{3\Delta}\right)\left(\frac{3^{-3/4}e^{\frac{2\pi\sqrt{\Delta}}{\sqrt{3}}}}{4\Delta^{5/4}}\right). \tag{29}$$

## 2.2 Analysis via integer partitions and the Hardy-Ramanujan formula

We conclude this section by noting that the $q^{1/24}\eta^{-1}$ is the generating function for the number of partitions of an integer[3] i.e we have

$$PE(q,\bar{q}) = \sum_n P(n)q^n\sum_m P(m)\bar{q}^m, \tag{30}$$

---

[3]TM acknowledges Brian Henning in communicating formulas and for helpful discussions on the results of this section.

where $P(n)$ is the number of partitions of an integer $n$[4]. The generating function for scalars (without IBP) is obtained by picking out terms from the above that have the same powers of $q$ and $\bar{q}$:

$$H(q) = \sum_n P(n)^2 q^{2n} = \sum_\Delta P\left(\frac{\Delta}{2}\right)^2 q^\Delta. \tag{31}$$

If we impose IBP, then we have

$$PE_{\text{IBP}}(q,\bar{q}) = (1-q)(1-\bar{q})PE(q,\bar{q}) = \sum_n (P(n)-P(n-1))q^n \sum_m (P(m)-P(m-1))\bar{q}^m, \tag{32}$$

and the generating function for scalars with IBP imposed is given by

$$H_{\text{IBP}}(q) = \sum_n (P(n)-P(n-1))^2 q^{2n}. \tag{33}$$

Thus one can relate the asymptotics with asymptotic growth of number of partition $P(n)$ of integer $n$. The asymptotic $n \to \infty$ limit of $P(n)$ is given by the famous Hardy-Ramanujan formula [44]

$$P(n) \underset{n\to\infty}{\simeq} \frac{e^{2\pi\sqrt{\frac{n}{6}}}}{4\sqrt{3}n}. \tag{34}$$

In particular, we have

$$\begin{aligned} \rho^{\text{scalar}}(\Delta) &= \left[P\left(\frac{\Delta}{2}\right)\right]^2, \\ \rho^{\text{scalar}}_{\text{IBP}}(\Delta) &= \left[P\left(\frac{\Delta}{2}\right)-P\left(\frac{\Delta}{2}-1\right)\right]^2 \underset{\Delta\to\infty}{\simeq} \left(\frac{dP(n)}{dn}\right)^2\bigg|_{n=\Delta/2}, \end{aligned} \tag{35}$$

where it is to be understood that we are taking a derivative of the function appearing on the R.H.S of Eq. (34). It is easily verified that Eq. (23) and Eq. (29) are reproduced.

# 3 A new trick

Although the above calculations are clean, they are not directly applicable in higher dimensions where we do not have the luxury of having a modular covariant function in the PE. In this section we will step down on rigour a bit and invent a new way to obtain asymptotics up to $O(1)$ terms. In the following section we will show how to regain these $O(1)$ terms, but we choose to present the following trick first, as *(i)* it is a fast and transparent way to obtain the form of the asymptotic growth, and *(ii)* we will later use this trick—again for its simplicity of presentation—to analyse projections of the PE to obtain Hilbert series, appealing at the end to the rigorous treatment to fix the $O(1)$ terms.

## 3.1 Reproducing $d = 2$ asymptotics

Let us first demonstrate this trick for a single scalar field in $d = 2$ so as to make direct comparison with what we obtained in the previous section. We start with the regularized PE of Eq. (12),

$$PE = \exp\left[\sum_{n=1}^\infty \frac{1}{n}\frac{q^n}{1-q^n}\right]\exp\left[\sum_{n=1}^\infty \frac{1}{n}\frac{\bar{q}^n}{1-\bar{q}^n}\right]. \tag{36}$$

---

[4]This is to be distinguished from the plane partition that we discuss later in §3.2

**All operators**

We set $q = \bar{q} = e^{-\beta}$ for the analysis of all operators. Then we have

$$PE = \exp\left[\sum_{n=1}^{\infty} \frac{e^{-\beta n}}{n} \frac{2}{1 - e^{-n\beta}}\right].\tag{37}$$

Now we do the following series expansion for small $\beta$,

$$\frac{2}{1 - e^{-n\beta}} = \frac{2}{\beta n} + 1 + \frac{\beta}{6} - \frac{\beta^3 n^3}{360} + \frac{\beta^5 n^5}{15120} + \cdots,\tag{38}$$

take the right hand side of the above and do the summation

$$\exp\left[\sum_{n=1}^{\infty} \frac{e^{-n\beta}}{n}\left[\underbrace{\frac{2}{\beta n} + 1}_{\text{singular pieces}} + \frac{\beta n}{6} - \frac{\beta^3 n^3}{360} + \frac{\beta^5 n^5}{15120}\right]\right],\tag{39}$$

and take the $\beta \to 0$ limit. The underbracket labelled "singular pieces" refers to the fact that these will eventually produce the singular pieces in the $\log PE$ in the $\beta \to 0$ limit.

The singular pieces indicated in (39), after summing over $n$, give an exponent $\frac{\pi^2}{3\beta} + \log(\beta) - 2$. This is very close to the actual value $\frac{\pi^2}{3\beta} + \log(\beta) - \log(2\pi)$ as in Eq. (16). The non-rigorous part of this analysis is we are not proving that the non-singular terms we threw away in Eq. (38) changes the order one multiplicative prefactor in the $\beta \to 0$ asymptotics of the PE (not to be confused with the leading term in $\log PE$, which we know exactly). Nonetheless, we do reproduce Eq. (16) up to an order one multiplicative correction,

$$PE(q = \bar{q} \to 1) \simeq \beta e^{-2} \exp\left[\frac{\pi^2}{3\beta}\right].\tag{40}$$

We can compare the ratio of actual asymptotics and the one we just obtained:

$$\text{actual asymptotics}_{(16)} : \text{asymptotics via new trick}_{(40)} = \frac{1}{2\pi} : e^{-2} = 1 : 0.85.\tag{41}$$

The non-singular pieces in Eq. (38) produce 0 if we take the $\beta \to 0$ limit first and then sum over $n$; on the other hand, if we first sum over $n$ and then take the $\beta \to 0$ limit, they produce $\beta$ independent order one corrections. If we keep up to $\beta^5$ term, sum over $n$ and then take the $\beta \to 0$ limit, we find

$$\text{actual asymptotics}_{(16)} : \text{asymptotics via new trick}_{\text{modified }(40)} = \frac{1}{2\pi} : e^{-\frac{463}{252}} = 1 : 1.000575645.\tag{42}$$

Unfortunately, we can not improve the order one number by keeping more and more higher order terms—at some point the non-rigourous nature of our analysis ensures that the ratio will become far off from 1. However, we will use this trick on the principle that the singular pieces (and perhaps a small number of higher order terms) reproduce the correct asymptotics up to an order one number that is reasonably close to unity. Note that once we have the asymptotics of $PE(\beta)$, we can again use the inverse Laplace transformation to re-obtain Eq. (17), again up to order one multiplicative terms.

**Projecting onto scalars**

To apply the above trick to the asymptotic growth of scalar operators, we reintroduce the fugacity for spin and write

$$PE(\beta, \omega) = \exp\left[\sum_{n=1}^{\infty} \frac{e^{-n\beta}}{n} f(\beta, \omega)\right], \tag{43}$$

where we have

$$f(\beta, \omega) \equiv \frac{e^{-2\pi\iota n\omega}}{1 - e^{-n(\beta + 2\pi\iota\omega)}} + \frac{e^{2\pi\iota n\omega}}{1 - e^{-n(\beta - 2\pi\iota\omega)}}.$$

There is a saddle at $\omega = 0$, so we first expand $f$ around this point, keeping terms up to order $\omega^2$,

$$f(\beta, \omega) \simeq \left[\frac{2}{1 - e^{-\beta n}}\right] - 4\pi^2 n^2 \omega^2 \left[\frac{e^{2\beta n}(e^{\beta n} + 1)}{(e^{\beta n} - 1)^3}\right] + \cdots. \tag{44}$$

We take the $\beta \to 0$ limit of the terms appearing in the square brackets of the above expression

$$f(\beta, \omega) \simeq \left[1 + \frac{2}{\beta n}\right] - 4\pi^2 n^2 \omega^2 \left[\frac{2}{\beta^3 n^2} + \frac{2}{n\beta^2} + \frac{1}{\beta} + \frac{n}{3}\right]. \tag{45}$$

The fluctuation around the saddle is controlled by the term proportional to $\omega^2$. The most singular term in $\beta \to 0$ limit that is present in this term is proportional to $\beta^{-3}$; we keep this term only and plug the truncated $f(\beta, \omega)$ back into the expression for $PE(\beta, \omega)$ in Eq. (43). Subsequently, we perform the sum over $n$ and again take $\beta \to 0$ limit,

$$\begin{aligned} PE(\beta, \omega) &= \sum_{n=1}^{\infty} e^{-n\beta} \left[\frac{2}{\beta n^2} + \frac{1}{n}\right] - \omega^2 \left[\frac{8\pi^2}{\beta^3 n^2}\right] \\ &\underset{\beta \to 0}{=} \frac{\pi^2}{3\beta} - 2 + \log(\beta) - \frac{4(\pi^4 \omega^2)}{3\beta^3}. \end{aligned} \tag{46}$$

Finally we do the projection by integrating over the Haar measure,

$$\begin{aligned} H(\beta) &\underset{\beta \to 0}{\simeq} \beta e^{-2} \exp\left[\frac{\pi^2}{3\beta}\right] \int_{-1/2}^{1/2} d\omega \, \exp\left[-\frac{4(\pi^4 \omega^2)}{3\beta^3}\right] \\ &\simeq \beta e^{-2} \exp\left[\frac{\pi^2}{3\beta}\right] \int_{-\infty}^{\infty} d\omega \, \exp\left[-\frac{4(\pi^4 \omega^2)}{3\beta^3}\right] = \frac{\sqrt{3}\beta^{5/2}}{2\pi^{3/2}} e^{\frac{\pi^2}{3\beta} - 2}, \end{aligned} \tag{47}$$

where we have used the saddle point approximation and enlarged the integration region from $-\infty$ to $+\infty$ to integrate over the Gaussian fluctuation around the saddle. This precisely reproduces Eq. (22) up to order one multiplicative correction.

**Imposing IBP**

Imposing IBP requires us to multiply the PE in Eq. (40) or $H$ in Eq. (47) by $\beta^2$. They match onto Eq. (26) and Eq. (28) respecctively upto order one multiplicative correction.

## 3.2 Plane partitions

In this subsection, we proceed to test our new trick on the plane partitions of an integer and relate the generating function for plane partitions to the $d = 3$ Hilbert series for scalar field theory.

A plane partition is defined to be a two dimensional array of non-negative integers $\{a_{i,j}\}_{i,j\in\mathbb{Z}_{\geq 0}}$ such that

$$a_{i,j} \geq a_{i+1,j} \qquad a_{i,j} \geq a_{i,j+1}. \tag{48}$$

The sum of a plane partition is defined as

$$n = \sum_{i,j} a_{i,j}. \tag{49}$$

Let us define $PL(n)$ to be the number of plane partitions of integer $n$. For example, for $n = 3$ we have 6 partitions,

$$\begin{bmatrix} 1 \\ 1 \\ 1 \end{bmatrix}, \begin{bmatrix} 1 & 1 \\ 1 & \end{bmatrix}, \begin{bmatrix} 1 & 1 & 1 \end{bmatrix}, \begin{bmatrix} 2 \\ 1 \end{bmatrix}, \begin{bmatrix} 2 & 1 \end{bmatrix}, \begin{bmatrix} 3 \end{bmatrix}. \tag{50}$$

The generating function for $PL(n)$ is given by [45]

$$PE_{pl}(q) \equiv \sum_n PL(n) q^n = \prod_{n=1}^{\infty} (1 - q^n)^{-n} = \exp\left[ \sum_{n=1}^{\infty} \frac{q^n}{n} \frac{1}{(1-q^n)^2} \right]. \tag{51}$$

Again, in the following we will use the variable $q = e^{-\beta}$. The asymptotic growth of $PL(n)$ is given by the inverse Laplace transformation of the $\beta \to 0$ limit of

$$PE_{pl}(\beta) = \exp\left[ \sum_{n=1}^{\infty} \frac{e^{-n\beta}}{n} \frac{1}{(1-e^{-n\beta})^2} \right]. \tag{52}$$

We apply our trick (keeping the singular pieces before summing over $n$ i.e. $(1 - e^{-n\beta})^{-2} \simeq \frac{1}{\beta^2 n^2} + \frac{1}{\beta n} + \frac{5}{12}$). This produces

$$PE_{pl}(\beta) \underset{\beta \to 0}{\simeq} e^{-1/4} \beta^{1/12} \exp\left[ \frac{\zeta(3)}{\beta^2} \right], \tag{53}$$

which upon inverse Laplace transformation gives

$$PL(n) \underset{n \to \infty}{\simeq} \frac{\zeta(3)^{7/36}}{2^{11/36}\sqrt{3\pi} n^{25/36}} \exp\left[ 3 \times 2^{-2/3} n^{2/3} (\zeta(3))^{1/3} - 1/4 \right]. \tag{54}$$

The asymptotic growth of $PL(n)$ is known in the mathematics literature: it was first figured out in an old paper by Wright [45], and a typographical error in Wright's paper was pointed out later by Mutafchiev and Kamenov [46]. The growth is given by

$$PL(n) \underset{n \to \infty}{\simeq} \frac{\zeta(3)^{7/36}}{2^{11/36}\sqrt{3\pi} n^{25/36}} \exp\left[ 3 \times 2^{-2/3} n^{2/3} (\zeta(3))^{1/3} + \zeta'(-1) \right]. \tag{55}$$

Comparing Eq. (55) and Eq. (54) we have

$$\text{actual asymptotics} : \text{asymptotics via new trick} = e^{\zeta'(-1)} : e^{-1/4} = 1 : 0.92. \tag{56}$$
$$\quad\quad\quad (55) \quad\quad\quad\quad\quad\quad\quad\quad (54)$$

The actual/obtained via trick asymptotics of $PL(n)$ can also be turned into actual/obtained via trick asymptotics of $PE_{pl}(\beta \to 0)$,

$$
PE_{pl}^{\text{Maths}}(\beta) \underset{\beta \to 0}{\simeq} \beta^{1/12} \exp\left[\frac{\zeta(3)}{\beta^2} + \zeta'(1)\right]
$$

$$
PE_{pl}^{\text{trick}}(\beta) \underset{\beta \to 0}{\simeq} \beta^{1/12} \exp\left[\frac{\zeta(3)}{\beta^2} - 1/4\right].
$$
(57)

The superscript in $PE$ signifies how we obtain the result. Again, if we keep up to the first order piece before summing over $n$ (i.e $(1 - e^{-n\beta})^{-2} \simeq \frac{1}{\beta^2 n^2} + \frac{1}{\beta n} + \frac{5}{12} + \frac{n\beta}{12}$), we find that we obtain a better result

$$
PE_{pl}^{\text{trick, first order}}(\beta) \underset{\beta \to 0}{\simeq} \beta^{1/12} \exp\left[\frac{\zeta(3)}{\beta^2} - 1/6\right]
$$

$$
\text{actual asymptotics : asymptotics via new trick} = e^{\zeta'(-1)} : e^{-1/6} = 1 : 0.99875.
$$
(58)

$$
\underset{(55)}{} \qquad \underset{(54)\ \text{incl. first order}}{}
$$

### 3.3  Relation of plane partitions to $d = 3$ Hilbert series

The PE for a scalar field theory in $d = 3$ can be massaged into a form that is directly related to the PE for plane partitions[5]. The steps to do so (that we spell out in detail at this point) are as follows where we start by considering the explicit form of PE from [28] by setting all the fugacities to 1:

$$
\begin{aligned}
PE(\beta) &= \exp\left[\sum_{n=1}^{\infty} \frac{e^{-n\beta/2}}{n} \frac{1 - e^{-2n\beta}}{(1 - e^{-n\beta})^3}\right] \\
&= \exp\left[\sum_{n=1}^{\infty} \frac{e^{-n\beta/2}}{n} \sum_{k=0}^{\infty} (2k+1)e^{-kn\beta}\right] \\
&= \exp\left[\sum_{k=0}^{\infty} (2k+1) \sum_{n=1}^{\infty} \frac{e^{-n\beta(k+1/2)}}{n}\right] \\
&= \prod_{k=0}^{\infty} \left(1 - q^{k+1/2}\right)^{-(2k+1)} = \prod_{n=0}^{\infty} \left(1 - \sqrt{q}^{2n+1}\right)^{-(2n+1)} = \frac{\prod_{n=1}^{\infty}\left(1 - \sqrt{q}^n\right)^{-n}}{\prod_{n=1}^{\infty}\left(1 - \sqrt{q}^{2n}\right)^{-(2n)}} \\
&= \frac{\prod_{n=1}^{\infty}\left(1 - \sqrt{q}^n\right)^{-n}}{\left[\prod_{n=1}^{\infty}(1 - q^n)^{-n}\right]^2} \\
&= \frac{PE_{pl}(\beta/2)}{\left[PE_{pl}(\beta)\right]^2}.
\end{aligned}
$$
(59)

In the last line, we relate it to (51) introduced previously as generating function of plane partition of integer. Now using the asymptotics for plane partitions of integer, we find that

$$
PE^{\text{Maths}}(\beta) \underset{\beta \to 0}{\simeq} 2^{-1/12} \beta^{-1/12} \exp\left[\frac{2\zeta(3)}{\beta^2} - \zeta'(-1)\right].
$$
(60)

The superscript in $PE$ signifies that we obtained the formula using Wright's result [45]. In comparison, our trick produces (keeping up to the first order term)

$$
PE^{\text{trick}}(\beta) \underset{\beta \to 0}{\simeq} 2^{1/12} \beta^{-1/12} \exp\left[\frac{2\zeta(3)}{\beta^2} + 1/24\right].
$$
(61)

---

[5]The plane partion also appears in the context of higher spin CFT in $2D$ [47, 48].

That is we find,

$$\text{actual asymptotics} : \text{asymptotics via new trick} = 2^{-1/12}e^{-\zeta'(-1)} : 2^{1/12}e^{1/24} = 1 : 0.99184.$$
$$\underset{(60)}{\qquad\qquad\qquad} \underset{(61)}{\qquad\qquad\qquad}$$

(62)

At this point, we expect the readers to be convinced that our trick produces the singular $\beta$ dependence correctly. Furthermore, we observe that can produce the order one number almost correctly, but not quite. In the following section we proceed to capture this order one number exactly. The hint for how to do so is in the proof of the asymptotics of plane partitions, which uses a theorem named after Meinardus.

# 4 Asymptotic growth of operator degeneracy from Meinardus' theorem

Our aim is to rigorously obtain the asymptotic growth of number of operators for EFTs in general spacetime dimension $d$, encompassing particles of general spin $j$ (that saturate the unitarity bounds). We show this can be achieved using the theory of partitions (the term 'partitions' here is not in the sense of a direct higher dimensional generalization of the partition of integers, as with the plane partitions above). In particular we appeal to a theorem due to Meinardus [49]. A mathematical exposition of this theorem can be found in the Chapter 6 of [50] (see Theorem 6.2).

Meinardus' theorem provides us with a technique to perform the two step process laid out in the Introduction: first, we need to know the $\beta \to 0$ behavior of the generating function; second, we do an inverse Laplace transform to deduce the asymptotic growth. The theorem itself pertains to the size of the error terms generated by these two steps. In what follows, we will employ the techniques used in the proof of the theorem, without keeping careful track of the error terms. This is because, while in some scenarios, the error terms could follow directly as per Meinardus, in most of the scenarios we consider (including any projection to particular singlet sectors), keeping track of the error is much more involved and beyond the scope of this paper. Thus, we refer the reader to the original reference for an explicit statement of the theorem, and proceed below by explaining the techniques we use through the example of counting of operators in Bosonic scalar field theory.

We will in fact need generalize some of the techniques needed for the proof for application to the general EFTs we are interested in; we will highlight these tricks we employ as they come up in the following. In fact, all of the necessary tricks to obtain asymptotic formulae for arbitrary $d$ and spin $j$ EFTs can be showcased by presenting the cases of scalars (bosonic Meinardus theorems) and spin half fermions (fermionic Meinardus theorems) in even dimensions. We will thus work through these cases here; explicit results in odd dimensions and with particles of higher spin $j$ are collected in App. B. However, the leading behaviour in arbitrary $d$ and $j$ is universal and compact, and we present this in Sec. 4.3 below.

Before proceeding, let us make a brief comment regarding Cardy's approach in [32]. The technique involved in that paper is related to Meinardus' theorem although Meinardus' theorem was not stated explicitly, nor leveraged to find results beyond leading order. With the techniques borrowed from Meinardus' theorem it is possible to go beyond leading order. Furthermore, in the following we apply the technique to study the growth of subsets of operators *e.g.* sitting in the singlet sector of some global symmetry group and/or Lorentz group.

## 4.1 Bosonic Meinardus theorems

The purpose of this subsection is to spell out the techniques involved in Meinardus theorem for the purpose of counting operators in Bosonic scalar field theory.

We begin by manipulating the PE for a scalar field theory in arbitrary even $d \geq 4$ dimension into the following form

$$
\begin{aligned}
PE(d, \beta) &= \exp\left[\sum_{n=1}^{\infty} \frac{e^{-n\frac{d-2}{2}\beta}}{n} \frac{1 - e^{-2n\beta}}{(1 - e^{-n\beta})^d}\right] \\
&= \exp\left[\sum_{n=1}^{\infty} \frac{e^{-n\frac{d-2}{2}\beta}}{n} \sum_{k=0}^{\infty} f(k, d) e^{-kn\beta}\right] \\
&= \prod_{k=0}^{\infty} \left(1 - q^{k+\frac{d-2}{2}}\right)^{-f(k,d)} = \prod_{k=1}^{\infty} \left(1 - q^k\right)^{-f(k-d/2+1,d)},
\end{aligned}
\tag{63}
$$

where $f(k, d)$ is given by the symmetric spin $k$ respresentation of $SO(d)$ i.e.

$$
f(k, d) = \dim[k, 0, 0, \cdots].
\tag{64}
$$

For example

$$
f(k, 4) = (k+1)^2, \quad f(k, 6) = \frac{1}{12}(k+1)(k+2)^2(k+3), \quad etc.
\tag{65}
$$

Note the shift in the variable $k$ in the final equality of Eq. (63) is valid because $f(k - d/2 + 1, d) = 0$ for $1 \leq k < (d-2)/2$. (For $d = 4$, the shift is trivially valid.)

The rational behind recasting the PE in a product form starting from $k = 1$ is that the Meinardus theorem deals with such infinite product. We proceed to sketch the proof of the theorem, and show how the leading behaviour of the PE, and hence the asymptotic growth of operators, can be obtained. The proof proceeds via two steps: we first figure out the $PE(d, \beta)$ in $\beta \to 0$ limit, and then translate the result into asymptotics of growth of $q$ expansion coefficients.

We start from

$$
PE(d, \beta) = \prod_{k=1}^{\infty} \left(1 - q^k\right)^{-f(k-d/2+1,d)} = \exp\left[\sum_{n=1}^{\infty} \frac{1}{n} \sum_{k=1}^{\infty} f(k-d/2+1, d) e^{-kn\beta}\right].
\tag{66}
$$

We replace $e^{-kn\beta}$ by the Mellin transform and interchage the sum and the Mellin integral using absolute convergence. Thus we have

$$
\log PE(d, \beta) = \frac{1}{2\pi\iota} \int ds\, \beta^{-s} \Gamma(s) \zeta(s+1) D(d, s),
\tag{67}
$$

where the function $D(d, s)$ is defined as

$$
D(d, s) \equiv \sum_{k=1}^{\infty} k^{-s} f(k-d/2+1, d).
\tag{68}
$$

Note the Mellin transform integral goes along a vertical line where the above sum converges.

### Four spacetime dimensions

Now let us focus on $d = 4$. In $d = 4$ the function evaluates to

$$D(4,s) = \zeta(s-2). \tag{69}$$

$D(4,s)$ converges for $Re(s) > 3$, and it admits an analytic continuation for $Re(s) > -C$ with $0 < C < 1$ except for a simple first order pole at $s = 3$ with residue 1. In general, we denote the pole of $D(d,s)$ as $\alpha$ and the residue as $A$. Here we have

$$\alpha = 3, \quad A = 1. \tag{70}$$

Using the analytic structure of $D(d,s)$ we rewrite Eq. (67) specifying the contour:

$$\log PE(d,\beta) = \frac{1}{2\pi\iota} \int_{1+\alpha-\iota\infty}^{1+\alpha+\iota\infty} ds\, \beta^{-s} \Gamma(s)\zeta(s+1)D(d,s). \tag{71}$$

Now the idea is to move the contour to the left, crossing past the pole to a vertical line $Re(s) = -C$. There are two obstacles to such a movement. Firstly, the integrand has a simple first order pole at $s = \alpha = 3$ coming from $D(4,s)$ with residue $A\Gamma(\alpha)\zeta(\alpha+1)\beta^{-\alpha}$. Secondly, there is a second order pole at $s = 0$. We pick up a contribution from both poles, resulting in

$$\log PE(d,\beta) = A\Gamma(\alpha)\zeta(\alpha+1)\beta^{-\alpha} - D(4,0)\log(\beta) + D'(4,0)$$
$$+ \underbrace{\frac{1}{2\pi\iota} \int_{-C-\iota\infty}^{-C+\iota\infty} ds\beta^{-s}\Gamma(s)\zeta(s+1)D(d,s)}_{\text{Error term}}. \tag{72}$$

After showing that $\frac{1}{2\pi\iota}\int_{-C-\iota\infty}^{-C+\iota\infty} ds\beta^{-s}\Gamma(s)\zeta(s+1)D(d,s)$ is indeed an error term, one arrives at (Lemma 6.1, Eq. 6.2.1 of [50]),

$$\log PE(4,\beta) \underset{\beta\to 0}{\simeq} A\Gamma(\alpha)\zeta(\alpha+1)\beta^{-\alpha} - D(4,0)\log(\beta) + D'(4,0) = 2\zeta(4)\beta^{-3} + \zeta'(-2). \tag{73}$$

The second part of the Meinardus theorem involves translating the above result in asymptotics of $\rho(\Delta)$; this is essentially done using the saddle point method and carefully estimating the error. Evaluating the inverse Laplace transform in the saddle approximation we find for a scalar field in $d = 4$,

$$\rho(\Delta) \underset{\Delta\to\infty}{\simeq} \frac{1}{2\sqrt{2}\sqrt[8]{15}\Delta^{5/8}} \exp\left(\frac{4\pi\Delta^{3/4}}{3\sqrt[4]{15}} + \zeta'(-2)\right). \tag{74}$$

In Fig. 1 we show the exact $\rho(\Delta)$ obtained from a Taylor series expansion of the PE, compared with the asymptotic formula Eq. (74). In the upper panel of the plot, the asymptotic curve is indistinguishable from the data. In the lower panel we show the error of the asymptotic result: even at lowest mass dimensions this error is less than ten percent, and it decreases below the per mille level at dimension $\Delta \simeq 500$.

Before moving on, we can compare the expression for the growth of $PE(d,\beta)$ in $\beta \to 0$ limit derived above to that obtained via the new trick of Sec. 3. The latter, (keeping up to first order term in $\beta$ before summing over $n$) yields

$$\log PE(4,\beta) \underset{\beta\to 0}{\simeq} = 2\zeta(4)\beta^{-3} - \frac{13}{360}. \tag{75}$$

Now we have

$$\text{actual asymptotics : asymptotics via new trick} = e^{-\zeta'(-2)} : e^{-13/360} = 1 : 0.935. \tag{76}$$
$$\underset{\exp[(73)]}{} \qquad\qquad\qquad \underset{\exp[(75)]}{}$$

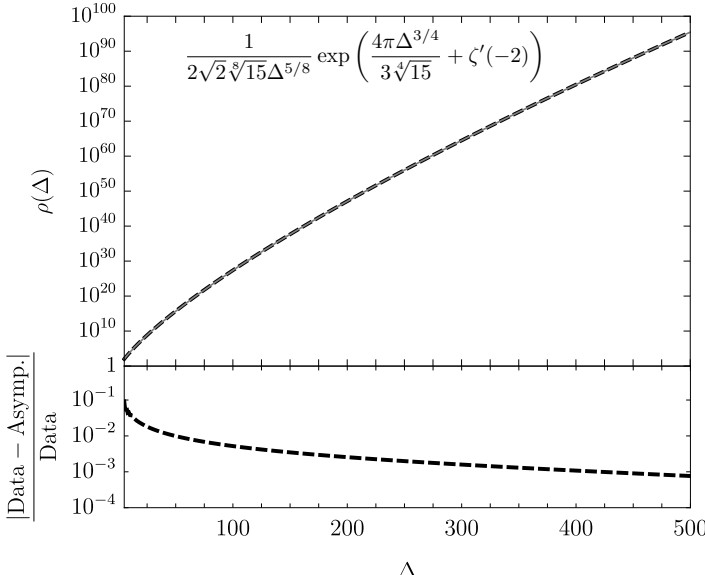

Figure 1: The growth of operators for a single real scalar in $d = 4$. *Upper panel:* The thick dashed line corresponds to exact data, obtained by a direct expansion of the PE. The thin grey curve that is indistinguishable from the data in the upper panel is the asymptotic result. *Lower panel:* Relative error between data and the asymptotic result.

## All even spacetime dimensions

We now proceed to analyse scalar field theory in $d = 2n$ dimensions for $n \geq 3$. Here we need a modified version of the theorem, which allows multiple but finite number of poles for the function $D(d, s)$ for $Re(s) > -C$ with $0 < C < 1$. Let us illustrate the concept in $d = 6$ and $d = 8$ dimensions. Starting from the general Hilbert series, we find, for example, in $d = 6, 8$, using Eq. (68)

$$
\begin{aligned}
D(6, s) &= \frac{1}{12} \left[ \zeta(s-4) - \zeta(s-2) \right] \\
D(8, s) &= \frac{1}{360} \left[ \zeta(s-6) - 5\zeta(s-4) + 4\zeta(s-2) \right].
\end{aligned}
\tag{77}
$$

Now these functions converge for $Re(s) > 5, 7$ and their analytic continuation have poles at $\alpha_i(d)$, given by

$$
\begin{aligned}
\alpha_1(6) &= 5, \quad \alpha_2(6) = 3 \\
\alpha_1(8) &= 7, \quad \alpha_2(8) = 5 \quad \alpha_3(8) = 3.
\end{aligned}
\tag{78}
$$

The corresponding residues are given by

$$
\begin{aligned}
A_1(6) &= 1/12, \quad A_2(6) = -1/12 \\
A_1(8) &= 1/360, \quad A_2(8) = -1/72 \quad A_3(8) = 1/90.
\end{aligned}
\tag{79}
$$

The modified Lemma 6.1 reads now

$$
\log PE(\beta) \underset{\beta \to 0}{\simeq} \sum_i A_i \Gamma(\alpha_i) \zeta(\alpha_i + 1) \beta^{-\alpha_i} - D(d, 0) \log(\beta) + D'(d, 0),
\tag{80}
$$

where the sum over poles come from shifting the defining contour from $Re(s) = 1 + \alpha_1$ to $Re(s) = -C$. For $d = 6, 8$ we have

$$
\begin{aligned}
\log PE^{d=6}(\beta) &\underset{\beta \to 0}{\simeq} \frac{2\pi^6}{945\beta^5} - \frac{\pi^4}{540\beta^3} + \frac{1}{12}\left[\zeta'(-4) - \zeta'(-2)\right] \\
\log PE^{d=8}(\beta) &\underset{\beta \to 0}{\simeq} \frac{\pi^8}{4725\beta^7} - \frac{\pi^6}{2835\beta^5} + \frac{\pi^4}{4050\beta^3} + \frac{1}{360}\left[\zeta'(-6) - 5\zeta'(-4) + 4\zeta'(-2)\right].
\end{aligned}
\tag{81}
$$

The results obtained from our trick (keeping up to first order term before summing over $n$) is

$$
\begin{aligned}
\log PE^{d=6}(\beta) &\underset{\beta \to 0}{\simeq} \frac{2\pi^6}{945\beta^5} - \frac{\pi^4}{540\beta^3} + \frac{379}{302400} \\
\log PE^{d=8}(\beta) &\underset{\beta \to 0}{\simeq} \frac{\pi^8}{4725\beta^7} - \frac{\pi^6}{2835\beta^5} + \frac{\pi^4}{4050\beta^3} - \frac{52729}{38102400}.
\end{aligned}
\tag{82}
$$

In comparison, we have for $d = 6$

$$
\underset{\underset{\exp[(81)]}{}}{\text{actual asymptotics}} : \underset{\underset{\exp[(82)]}{}}{\text{asymptotics via new trick}} = e^{\frac{1}{12}\left[\zeta'(-4)-\zeta'(-2)\right]} : e^{\frac{379}{302400}} = 1 : 0.998, \tag{83}
$$

and for $d = 8$

$$
\underset{\underset{\exp[(81)]}{}}{\text{actual asymptotics}} : \underset{\underset{\exp[Eq.\ (82)]}{}}{\text{asymptotics via new trick}}
$$
$$
= e^{\frac{1}{360}\left[\zeta'(-6)-5\zeta'(-4)+4\zeta'(-2)\right]} : e^{-\frac{52729}{38102400}} = 1 : 0.999. \tag{84}
$$

### 4.2 Fermionic Meinardus theorems

For the fermionic Meinardus theorem, we follow similar steps as in the scalar case to massage the PE to the following form:

$$
PE^{(f)}(\beta) = \prod_{n=0}^{\infty} \left(1 + q^{n+\frac{d-1}{2}}\right)^{g(n,d)}, \tag{85}
$$

where $g(n, d)$ is given by the dimension of the following representation of $SO(d)$:

$$
g(n, d) = \dim\left[n + 1/2, 1/2, \cdots\right]. \tag{86}
$$

We focus here on even spacetime dimensions and we have

$$
g(n, 4) = (n+1)(n+2), \quad g(n, 6) = \frac{1}{6}(1+n)(2+n)(3+n)(4+n) \quad \textit{etc.}
$$

In even dimensions, the canonical dimension of a spin $1/2$ fermion is half-integer. Thus we can not directly apply the Meinardus theorem in its original form. Another modification is needed.

We recast the PE in the following way

$$
PE^{(f)}(\beta) = \prod_{k=0}^{\infty}\left(1 + q^{n+\frac{d-1}{2}}\right)^{g(n,d)} = \prod_{k=1}^{\infty}\left(1 + q^{k-\frac{1}{2}}\right)^{g(k-d/2,d)} = \frac{\prod_{k=1}^{\infty}\left(1 + (\sqrt{q})^k\right)^{\tilde{g}(k,d)}}{\prod_{k=1}^{\infty}\left(1 + (\sqrt{q})^{2k}\right)^{\tilde{g}(2k,d)}}, \tag{87}
$$

where we have defined

$$
\tilde{g}(k, d) = g\left(k/2 + 1/2 - \frac{d}{2}, d\right).
$$

Again, in the first equality the shift in the variable $k$ is valid using the fact $g(k-d/2,d)=0$ for $1 \leq k < d/2$ and $k \in \mathbb{Z}_+$. The function $\tilde{g}$ is constructed in a way such that $\tilde{g}(2k-1,d)=g\left(k-1/2-\frac{d}{2},d\right)$ and the last equality follows. Thus the PE can be written as a ratio of two auxiliary PEs that are more amenable to the application of Meinardus' theorem:

$$PE^{(f)}(\beta) = \frac{PE^{\mathrm{aux.1}}(\beta/2)}{PE^{\mathrm{aux.2}}(\beta)}, \tag{88}$$

where the auxiliary PEs are given by

$$PE^{\mathrm{aux.1}}(\beta) = \prod_{k=1}^{\infty}\left(1+e^{-k\beta}\right)^{g(k/2+1/2-\frac{d}{2},d)},$$
$$PE^{\mathrm{aux.2}}(\beta) = \prod_{k=1}^{\infty}\left(1+e^{-k\beta}\right)^{g(k+1/2-\frac{d}{2},d)}. \tag{89}$$

We again define $D$ functions for each of these PEs,

$$D_1(d,s) = \sum_{k=1}^{\infty} k^{-s} g(k/2+1/2-\frac{d}{2},d),$$
$$D_2(d,s) = \sum_{k=1}^{\infty} k^{-s} g(k+1/2-\frac{d}{2},d). \tag{90}$$

Let us denote the poles of these functions as $\alpha_{i,1}$ and $\alpha_{j,2}$ with residues $A_{i,1}$ and $A_{j,2}$, where the second index $1,2$ refers to $D_1$ and $D_2$.

The analogue of Eq. (71) will read

$$\log PE^{\mathrm{aux.}}(d,\beta) = \frac{1}{2\pi\iota}\int_{1+\alpha-\iota\infty}^{1+\alpha+\iota\infty} ds\,\beta^{-s}\Gamma(s)(1-2^{-s})\zeta(s+1)D(d,s), \tag{91}$$

where the extra factor (compared to the Bosonic case) comes due to Fermionic nature of the PE; this factor changes the behaviour of the integrand around $s=0$—In particular, we don't obtain any $\log\beta$ piece. This is the final modification of Meinardus' theorem we will need. We find

$$\log PE^{(f)}(\beta \to 0) = \log PE^{\mathrm{aux.1}}(\beta/2 \to 0) - \log PE^{\mathrm{aux.2}}(\beta \to 0)$$
$$= \left(\sum_i A_{i,1}\Gamma(\alpha_{i,1})\left(1-2^{-\alpha_{i,1}}\right)\zeta(\alpha_{i,1}+1)(\beta/2)^{-\alpha_{i,1}}\right)$$
$$- \left(\sum_j A_{j,2}\Gamma(\alpha_{j,2})\left(1-2^{-\alpha_{j,2}}\right)\zeta(\alpha_{j,2}+1)\beta^{-\alpha_{j,2}}\right) + [D_1(d,0)-D_2(d,0)]\log(2). \tag{92}$$

For example, in $d=4$ we have

$$D_1(4,s) = \sum k^{-s} g\left(k/2+1/2-\frac{d}{2},d\right) = \frac{1}{4}[\zeta(s-2)-\zeta(s)]$$
$$D_2(4,s) = \sum k^{-s} g\left(k+1/2-\frac{d}{2},d\right) = \zeta(s-2)-\frac{1}{4}\zeta(s). \tag{93}$$

We have poles at $\alpha_1 = 3$ and $\alpha_2 = 1$ for both of the PE. The residues are different

$$A_{1,1} = -A_{2,1} = \frac{1}{4}, \quad A_{1,2} = 1, A_{2,2} = -\frac{1}{4}. \tag{94}$$

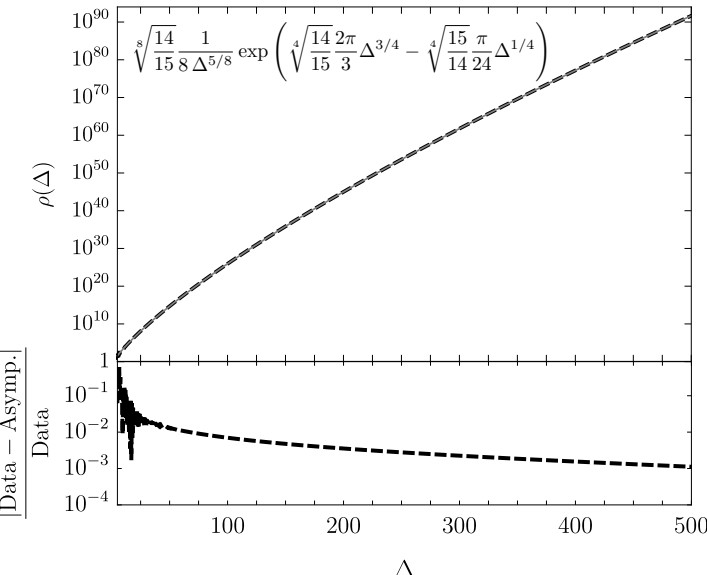

Figure 2: The growth of operators for a fermion in $d = 4$. *Upper panel:* The thick dashed line corresponds to exact data, obtained by a direct expansion of the PE. The thin grey curve that is indistinguishable from the data in the upper panel is the asymptotic result. *Lower panel:* Relative error between data and the asymptotic result.

Hence, we have using Eq. (92)

$$\log PE(\beta \to 0) = \frac{7}{4}\zeta(4)\beta^{-3} - \frac{1}{8}\zeta(2)\beta^{-1} = \frac{7\pi^4}{360}\beta^{-3} - \frac{\pi^2}{48}\beta^{-1}. \tag{95}$$

It is readily checked that this matches exactly with the result obtained from our trick (in fact, independent of the number of higher order terms one keeps before summing over $n$).

Performing the inverse Laplace transform, one obtains the asymptotic growth for the all operators in the fermionic PE in $d = 4$,

$$\rho(\Delta) \underset{\Delta \to \infty}{\simeq} \sqrt[8]{\frac{14}{15}} \frac{1}{4\Delta^{5/8}} \exp\left( \sqrt[4]{\frac{14}{15}} \frac{2\pi}{3} \Delta^{3/4} - \sqrt[4]{\frac{15}{14}} \frac{\pi}{24} \Delta^{1/4} \right). \tag{96}$$

In Fig. 2, the asymptotic formula above is compared with the exact values (data) from expanding the fermionic PE in $d = 4$. The asymptotic result in fact 'over counts' the number of operators by a factor of two at a fixed mass dimension, and the plotted asymptotic curve in Fig. 2 includes a factor of a half. We can understand this using intuition from plethora of examples in 2D CFT which says that the asymptotic formula is a count of the number operators lying in a window of $\delta = 1/2$, centred at $\Delta$, barring one of the end points if both the end points have operators. Similar results appeared in the context of the number of partitions of integers in [51], and are used in appendix B of [41]. We return to this point in the discussion section below. For now we simply reflect on the very good agreement between data and asymptotic expression shown in Fig. 2: as for the case of the scalar field, is at the level of ten percent at very low mass dimension, and decreases below the per mille level at dimension $\Delta \simeq 500$.

### 4.3 Leading behaviour in arbitrary dimension

The hilbert series in $\beta \to 0$ limit has the following form:

$$H(\beta \to 0) = \exp\left[\sum_{k=0}^{d-1} a_k \beta^{-k} + b\log(\beta) + c\right], \quad a_{d-1} > 0.$$

The leading singularity $a_{d-1}\beta^{-(d-1)}$ comes from the residue of the rightmost pole of the function $D(d,s)$. In even dimension, the residue of the rightmost pole is determined by the leading term in $D(s)$. We consider massless fields of spin $j$ that satisfy the unitarity bound. We recall the definition of $D(s)$:

$$D(s) = \sum k^{-s} g(k - k_*), \quad g(k) = \dim\left[k + j, \underbrace{j, j \cdots, j}_{d/2-1 \text{ times}}\right], \tag{97}$$

where $k_*$ is some constant shift depending on the canonical dimension of the field. Now the right most pole in $D(s)$ comes from the large $k$ piece of $g(k)$. We use the identity

$$g(k) = \dim\left[k + j, \underbrace{j, j, \cdots, j}_{d/2-1 \text{ times}}\right] \underset{k\to\infty}{\simeq} \frac{2}{\Gamma(d-1)} \dim\underbrace{[j, j, \cdots, j]}_{(d/2-1) \text{ times}}, \tag{98}$$

to extract the leading piece

$$D(d,s) \ni \sum_{k=1}^{\infty} k^{-s+d-2} \frac{2}{\Gamma(d-1)} \dim\underbrace{[j, j, \cdots, j]}_{(d/2-1) \text{ times}} = \frac{2f(r,j)}{\Gamma(d-1)} \zeta(s - d + 2), \tag{99}$$

where we have defined $f(r,j)$ as

$$f(r,j) \equiv \dim\underbrace{[j, j, \cdots, j]}_{(d/2-1) \text{ times}}. \tag{100}$$

One can immediately identify $f(r,j)$ with the dimension of spin $[j, j, \cdots, j]$ representation of the little group $SO(d-2)$ with $r$ being the rank of the group $SO(d)$. A few explicit forms are given below for low numerical values of $r$ as a function of $s$:

$$\begin{aligned}
f(2,j) &= 1 \\
f(3,j) &= (2j+1) \\
f(4,j) &= \frac{1}{6}(2j+3)(2j+2)(2j+1) \\
f(5,j) &= \frac{1}{360}(2j+1)(2j+2)(2j+3)^2(2j+4)(2j+5).
\end{aligned} \tag{101}$$

The leading singularity of $D(d,s)$ appears at $s = d-1$. Using the leading behavior of $f(r,j)$ we find that the residue at the leading singularity of $D(d,s)$ is given by $\frac{2f(r,j)}{\Gamma(d-1)}$. As a result, in $\beta \to 0$ limit, we have

$$\log H(\beta) \underset{\beta\to 0}{\simeq} 2f(r,j)\chi(\text{statistics})\zeta(d)\beta^{d-1} + O(\beta^{d-2}). \tag{102}$$

Here, $\chi(\text{statistic})$ is given by

$$\chi(\text{statistic}) = \begin{cases} 1, & \text{for Bosonic} \\ 1 - 2^{-d+1}, & \text{for Fermionic} \end{cases}. \tag{103}$$

The half integer $j$ in even dimension can be handled in a similar way. Furthermore, the function $f(r, j)\chi$(statistic) is additive if we have multiple fields.

For odd dimension, the only free fields that saturate the unitarity bounds are scalars and spin 1/2 fermions [52]. Now $f(r, j)$ becomes $f_{\text{odd}}(r, j)$ defined as

$$f_{\text{odd}}(r, 0) = 1 \,\&\, f_{\text{odd}}(r, 1/2) = 2^{r-1},$$

where $r = \frac{d-1}{2}$.

## 5 Hilbert Series Projections: Spin, internal symmetry, and IBP

In $d = 2$, we have seen that the effect of projecting onto the spin zero sector suppresses operator growth by a factor of $\beta^{3/2}$. The introduction of IBP further suppresses this growth by a factor of $\beta^2$. Here we will see how things work out in arbitrary dimension. We will also introduce internal symmetry and project onto singlets of these groups too. We find it more useful to do a concrete example in $d = 4$, rather than making everything very general and abstract. The idea is to watch out for the patterns in the simple $d = 4$ calculation and deduce the results for arbitrary $d$ dimension.

### 5.1 A worked example in $d = 4$

For our example, we consider complex scalars $\phi$ and $\phi^\dagger$ (charges $q_b$ and $-q_b$) and spin 1/2 fermions $\psi$ and $\psi^\dagger$ (charges $q_f$ and $-q_f$) There are some subtleties that lead to a factor of two if the theory contains only bosonic degree of freedom that we will discuss in the following section.

We will be using the $SU(2)_L \times SU(2)_R$ language for the fugacities $\alpha$ and $\gamma$ of the Lorentz group as in e.g. [3]. The Haar measure is

$$\int d\mu_{\text{Lorentz}} = \int d\mu_{SU(2)_L}(\alpha) \int d\mu_{SU(2)_R}(\gamma), \tag{104}$$

where e.g.

$$\int d\mu_{SU(2)_L} = \oint_{|\alpha|=1} \frac{d\alpha}{\alpha}(1 - \alpha^2). \tag{105}$$

The PE for this example EFT, is given by

$$
\begin{aligned}
PE = \exp&\left[ \sum_{n=1}^{\infty} \frac{1}{n} \frac{q^n(1 - q^{2n})}{P(q^n, \alpha^n, \gamma^n)} \left( \chi_{U(1)}(nq_b\omega) + \chi_{U(1)}(-nq_b\omega) \right) \right] \\
\times \exp&\left[ \sum_{n=1}^{\infty} (-1)^{n+1} \frac{1}{n} \frac{q^{\frac{3}{2}n}\left( \alpha^n + \alpha^{-n} - q^n(\gamma^n + \gamma^{-n}) \right)}{P(q^n, \alpha^n, \gamma^n)} \chi_{U(1)}(nq_f\omega) \right] \\
\times \exp&\left[ \sum_{n=1}^{\infty} (-1)^{n+1} \frac{1}{n} \frac{q^{\frac{3}{2}n}\left( \gamma^n + \gamma^{-n} - q^n(\alpha^n + \alpha^{-n}) \right)}{P(q^n, \alpha^n, \gamma^n)} \chi_{U(1)}(-nq_f\omega) \right],
\end{aligned}
\tag{106}
$$

where the character of the $U(1)$ internal symmetry is, in terms of the angular fugacity $w$,

$$\chi_{U(1)}(\omega) = e^{2\pi \iota \omega},$$

and where the momentum generating function is

$$P(q, \alpha, \gamma) = (1 - q \alpha \gamma)(1 - q \alpha \gamma^{-1})(1 - q \alpha^{-1} \gamma)(1 - q \gamma^{-1} \alpha^{-1}). \tag{107}$$

In the following we will work with fugacities $\beta$ and $\omega_{1,2}$ defined as

$$q = e^{-\beta}, \quad \alpha = e^{2\pi \iota \omega_1}, \quad \gamma = e^{2\pi \iota \omega_2}.$$

Let us analyse the PE using our by now familiar trick from Sec. 3. First we expand in the $\beta \to 0$ limit as

$$
\begin{aligned}
\log(PE) = & \\
& \sum_n \frac{e^{-n\beta}}{n} \left( \frac{2}{\beta^3 n^3} + \frac{4}{\beta^2 n^2} + \frac{3}{\beta n} - \frac{7}{6} - \frac{16\pi^2(\omega_1^2 + \omega_2^2)}{\beta^5 n^3} \right) \left( \chi_{U(1)}(nq_b \omega) + \chi_{U(1)}(-nq_b \omega) \right) \\
& + \sum_n \frac{(-1)^{n+1} e^{-\frac{3}{2} n\beta}}{n} \left( \frac{2}{\beta^3 n^3} + \frac{3}{\beta^2 n^2} + \frac{2}{\beta n} + \frac{3}{4} - \frac{16\pi^2(\omega_1^2 + \omega_2^2)}{\beta^5 n^3} \right) \\
& \times \left( \chi_{U(1)}(nq_f \omega) + \chi_{U(1)}(-nq_f \omega) \right),
\end{aligned}
\tag{108}
$$

where we kept only the singular terms in $\beta$.

The saddle for the $\omega$ integral will be determined by the leading term proportional to $\beta^{-3}$ i.e on $2\beta^{-3} \left[ g_b(\omega) + g_f(\omega) \right]$ where $g_{b/f}$ are given by

$$
\begin{aligned}
g_b(\omega) &= \sum_n \frac{2\cos(2\pi n q_b \omega)}{n^4} = \frac{\pi^4}{45} - \frac{2\pi^4 q_b^2 \omega^2}{3} + \frac{4\pi^4 q_b^3 |\omega|^3}{3} - \frac{2\pi^4 q_b^4 \omega^4}{3} \\
g_f(\omega) &= \sum_n (-1)^{n+1} \frac{2\cos(2\pi n q_f \omega)}{n^4} = \frac{7\pi^4}{360} - \frac{\pi^4 q_f^2 \omega^2}{3} + \frac{2\pi^4 q_f^4 \omega^4}{3}.
\end{aligned}
\tag{109}
$$

Note this term is independent of the spin, depending only on statistics. We keep up to the quadratic piece and do the integral over $\omega$. We remark that the quadratic piece can be obtained by expanding $\chi_{U(1)}(nq_{b/f}\omega) + \chi_{U(1)}(-nq_{b/f}\omega)$ first and then summing over $n$, and often this is a more practical approach to extract the asymptotics rather doing the sum explicitly as in Eq. (109). However, the way we proceeded above reveals the full periodic nature of the PE in $\omega$. We will assume that $U(1)$ charge is quantized in units of some base charge, with some field having base charge, and hence we can simple rescale all charges such that the base charge is unity[6]—in this case, Eq. (109) shows that the saddle is indeed $\omega = 0$.

---

[6]If one worked with base charge of integer $k > 1$, then there would be $k$ saddles to consider. The result should be multiplied by $k$ because of this, but would be suppressed by a factor of $k$ coming from a $1/Q$ (where $Q^2 = \sum_i q_i^2$) after the projection to $U(1)$ singlets, see Eq. (115) below. If on the other hand the base charge was $1/k$, one should use the $k$-th cover of $U(1)$, receiving a suppression by a factor of $1/k$ in the measure; this would similarly be cancelled by the $1/Q$ after projection.

Expanding, and performing the sum on $n$ in the terms with the angular fugacities,

$$
\begin{aligned}
\log(PE) = & \\
& \sum_n \frac{e^{-n\beta}}{n}\left(\frac{2}{\beta^3 n^3} + \frac{4}{\beta^2 n^2} + \frac{3}{\beta n} - \frac{7}{6}\right)(\dim_b) \\
& - \frac{8\pi^2(\omega_1^2 + \omega_2^2)}{\beta^5}\left[\frac{\pi^4}{45}\right](\dim_b) \\
& + \sum_n \frac{(-1)^{n+1}e^{-\frac{3}{2}n\beta}}{n}\left(\frac{2}{\beta^3 n^3} + \frac{3}{\beta^2 n^2} + \frac{2}{\beta n} + \frac{3}{4}\right)(\dim_f) \\
& - \frac{8\pi^2(\omega_1^2 + \omega_2^2)}{\beta^5}\left[\frac{7\pi^4}{360}\right](\dim_f) \\
& + 2\beta^{-3}\left[-(\dim_b)\frac{\pi^4 q_b^2 \omega^2}{3} - (\dim_f)\frac{1}{2}\frac{\pi^4 q_f^2 \omega^2}{3}\right],
\end{aligned}
\tag{110}
$$

where we have instated factors of $\dim_b$ and $\dim_f$ so as to keep track of the contribution of bosonic and fermionic degrees of freedom in this example (where $\dim_b = \dim_f = 2$).

The fugacity independent pieces will be

$$
\log PE(\beta) \ni \left[A\beta^{-3} + B\beta^{-1}\right] + C\zeta'(-2),
\tag{111}
$$

where $A, B$ have the universal form already seen above for scalars and spin-half fermioins, namely

$$
\begin{aligned}
A &= \left(\frac{\pi^4}{45}\dim_b + \frac{7\pi^4}{360}\dim_f\right) \\
B &= -\left(\frac{\pi^2}{48}\dim_f\right).
\end{aligned}
\tag{112}
$$

The order one piece is not correctly obtained through the above, but it is the result derived from using the Meinardus theorem,

$$
C = \dim_b.
\tag{113}
$$

Projecting onto Lorentz scalars is done by integrating the above about the saddle with the Haar measure for $SU(2) \times SU(2)$,

$$
\begin{aligned}
& \int_{-\infty}^{\infty} d\omega_1 \mu_{SU(2)}(\omega_1) \int_{-\infty}^{\infty} d\omega_2 \mu_{SU(2)}(\omega_2) \exp\left[-\frac{8\pi^2(\omega_1^2 + \omega_2^2)}{\beta^5}\left(\frac{\pi^4}{45}\dim_b + \frac{7\pi^4}{360}\dim_f\right)\right] \\
& = \frac{91125\beta^{15}}{32\pi^{13}}\left(\dim_b + \frac{7}{8}\dim_f\right)^{-3}.
\end{aligned}
\tag{114}
$$

If we had multiple scalars and fermions, with charges $q_i$, the $\omega$ dependent piece always contains a $\frac{2}{\beta^3 n^4}n^2\omega^2$ term multiplied by a sum over charges,

$$
PE(\beta) \ni \exp\left[-\frac{2\pi^4\omega^2}{3\beta^3}\left(\sum_i^{\dim_b/2} q_{bi}^2 + \frac{1}{2}\sum_i^{\dim_f/2} q_{fi}^2\right)\right],
$$

in the above example we have $q_{b1} = -q_{b2} = q_b$ and $q_{f1} = -q_{f2} = q_f$ and this matches with Eq. (110).

Projecting onto $U(1)$ singlets is done in following way:

$$\int_{-\infty}^{\infty} d\omega \exp\left[-\frac{2\pi^4\omega^2}{3\beta^3}\left(\sum_i q_{bi}^2 + \frac{1}{2}\sum_i q_{fi}^2\right)\right] = \sqrt{\frac{3}{2\pi^3}}\frac{\beta^{3/2}}{Q}$$

$$Q^2 \equiv \left(\sum_i q_{bi}^2 + \frac{1}{2}\sum_i q_{fi}^2\right).$$

(115)

## 5.2 General lessons

### General Lessons in $d = 4$

We start by presenting the master formula for the asymptotic behaviour of an EFT in $d = 4$, including particles up to spin $j = 1$ and including IBP relations. It follows from results that appeared above (and in App. B.3 for the spin 1 terms), and application of the saddle point approximations described in the previous section to project out singlets of general internal symmetry groups, $G$. The formula reads,

$$H(\beta) \underset{\beta\to 0}{\simeq} \left[K\beta^{\frac{3}{2}\dim \mathfrak{g}}\right]\left[\frac{91125\beta^{15}}{32\pi^{13}}\left(\dim_B + \frac{7}{8}\dim_f\right)^{-3}\right]\left[\beta^4\right]$$

$$\times \exp\left[A\beta^{-3} + B\beta^{-1} + C\zeta'(-2) + D\log\left(\frac{\beta}{2\pi}\right) + \text{higher spin}>1\right],$$

(116)

where

$$A = \left(\frac{\pi^4}{45}\dim_B + \frac{7\pi^4}{360}\dim_f\right),$$

$$B = -\left(\frac{\pi^2}{48}\dim_{1/2} + \frac{\pi^2}{6}\dim_1\right),$$

$$C = \dim_B,$$

$$D = -\frac{1}{2}\dim_1.$$

(117)

Here $\dim_B$ counts bosonic degrees of freedom (dof), $\dim_f$ counts fermionic dof, $\dim_{1/2}$ counts spin-1/2 dof, $\dim_1$ counts spin-1 dof, and $\dim \mathfrak{g}$ counts the dimension of the internal symmetry group $G$. The exact value of $K$ can be determined case by case; it depends on the symmetry group at hand, and also on the representations of the fields transforming under the symmetry. If $\dim_f = 0$, the result should be multiplied by a factor of two (see following section)

Let us discuss each of the terms in Eq. (116), working backwards, from right to left.

- The leading piece (exponential term) comes from the counting of all degrees of freedom in the EFT, turning off fugacities characterizing spin and internal symmetry.

- Imposing IBP provides a suppression of $\beta^4$.

- Projecting onto Lorentz scalars provides a suppression of $\beta^{15}$, and an order one multiplicative piece that depends on the number of bosonic and fermionic dof.

- Projecting onto singlets of an internal symmetry group $G$ provides a suppression in $\beta$ that is dependent on the dimension of $G$, and an order one multiplicative piece $K$, described above.

**General Lessons in arbitrary** $d$

The below results also follow from a direct application of the above techniques

- The leading piece (exponential term) again comes from the counting of all degrees of freedom in the EFT, turning off fugacities characterizing spin and internal symmetry (see also Sec. 4.3).

- Imposing IBP provides a suppression of $\beta^d$.

- Projecting onto singlet under any Lie group induces suppresion by

$$(\beta)^{g(d,G)\times\dim \mathfrak{g}},$$

where $\mathfrak{g}$ is the dimension of the Lie algebra corresponding to Lie group $G$. The function $g(d,G)$ is given by

$$\begin{cases} g(d,G) = \frac{d+1}{2}, & \text{where } G = \text{Lorentz}, \\ g(d,G) = \frac{d-1}{2}, & \text{where } G = \text{Global symmetry}. \end{cases} \tag{118}$$

That is, we find

- Projecting onto Lorentz scalars implies a suppression by $\beta^{d(d^2-1)/4}$ (irrespective of the spin of the fields).

- Projecting on to gauge singlet implies suppression by $(\beta^{(d-1)/2})^\ell$, where the quantity $\ell$ can be extracted from the volume measure of the gauge group in the fugacity going to 1 limit. For a continuous Lie group $G$, this identifies $\ell = \dim \mathfrak{g}$ as the dimension of the Lie algebra.

The suppression by IBP projection and onto singlets of the Lorentz and internal symmetry groups is less important than sub-leading (larger than logarithmic in $\beta$) corrections in the exponent. The (unquantified) polynomial suppression of IBP projection and of projection onto singlets of the Lorentz was pointed out in [28].

## 6 Saddle subtleties with and without fermions

For a purely Bosonic theory, one needs to multiply the result presented Eq. (116) by a factor of 2. To be precise, if we view $H(\beta \to 0)$ as a function of $\dim_f$, we have

$$H(\beta \to 0, \dim_f = 0) = 2\left(\lim_{\dim_f \mapsto 0}\left[H(\beta \to 0, \dim_f)\right]\right). \tag{119}$$

The above signals phase transition-like behaviour, and arises because of the presence of extra saddles that appear in the limit. The symmetry reason behind such phenomenon is that the presence of fermions break the symmetry $(\omega_1, \omega_2) \mapsto (\omega_1 + \frac{1}{2}, \omega_2 + \frac{1}{2})$, i.e. when $\dim_f = 0$, there is a symmetry enhancement. In terms of $\alpha = e^{2\pi\iota\omega_1}$ and $\gamma = e^{2\pi\iota\omega_2}$, the symmetry is implemented by $(\alpha, \gamma) \mapsto (-\alpha, -\gamma)$. While the bosonic PE is symmetric under this transformation, the fermionic (or the ones with both bosons and fermions) ones are not.

In the $\alpha, \gamma$ variables there are two saddles to keep track of: when $\alpha = \gamma = \pm 1$. The apparent discontinuity in Eq. (119) comes about because for $\dim_f = 0$, we have degenerate saddles but when $\dim_f \neq 0$, the symmetry is broken, and so is the degeneracy. Consequently, one of the saddles gets suppressed. This phenomenon happens universally in all dimensions, see App. C for the details. Keeping both the saddles restore the continuity of $H(\beta)$ as a function of number of fermions. We explicitly show this in Sec. 6.2 for $d = 4$.

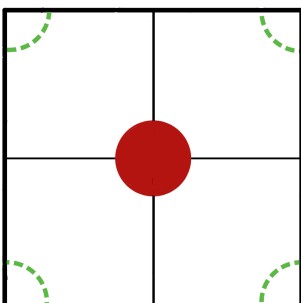

Figure 3: Calculation using $SU(2) \times SU(2)$. The figure depicts saddles on $(\omega_1, \omega_2)$ plane. The center of the square is at $(0,0)$. The corners are at $(1/2, 1/2), (1/2, -1/2), (-1/2, 1/2), (-1/2, 1/2)$. For the bosonic case, all the saddles contribute. When we add fermions in the mix, the green colored saddles at $(0, \pm 1/2), (\pm 1/2, 0)$ don't contribute anymore, leading to an overall factor of 1/2. We have drawn circular regions arounds the saddle to denote how much the fluctuation around the saddle contributes. For example, each of the corner ones contributes one quarter of the center one.

## 6.1 Double cover for bosons in $d = 4$

We begin by considering a single scalar field theory. The PE can either be written in the language of $SU(2) \times SU(2)$ or in the language of $SO(4)$. The purpose of this section is to clarify that the both will provide us with the same answer. If we include fermions in the theory, then we need to work with the double cover of $SO(4)$. This is is taken up in App. C, and generalized so as to studyng this phenomena in arbitrary $d$.

**Take I—The $SU(2) \times SU(2)$ way**

$H(\beta)$ is given by

$$H(\beta) = \oint d\alpha \oint d\gamma \left[ \frac{1}{\alpha\gamma}(1 - \alpha^2)(1 - \gamma^2) \right] \exp \left[ \sum_{n=1}^{\infty} \frac{e^{-n\beta}}{n} \frac{1 - e^{-2n\beta}}{P(e^{-n\beta}, \alpha^n, \gamma^n)} \right], \tag{120}$$

with $P$ defined in Eq. (107)

Now note that in the $\beta \to 0$ limit, the singularity appears when $\alpha\gamma = \alpha/\gamma = 1$. This admits two solutions

$$\alpha = \gamma = \pm 1. \tag{121}$$

The angular fugacities $\omega_i$ are defined as

$$\alpha = e^{2\pi\iota\omega_1}, \ \gamma = e^{2\pi\iota\omega_2}. \tag{122}$$

As $\alpha, \gamma$ traverse the circle once, we have a square region swept out in the $(\omega_1, \omega_2)$ plane with center at $(0,0)$ and vertices at $(\pm 1/2, \pm 1/2), (\pm 1/2, \mp 1/2)$. From Eq. (121) it follows that in the $(\omega_1, \omega_2)$ plane the saddles exist at the points (see fig. 3)

$$(\omega_1, \omega_2) = \{(0,0), (1/2, 1/2), (1/2, -1/2), (-1/2, 1/2), (-1/2, -1/2)\}. \tag{123}$$

We have already considered the $(0,0)$ saddle in the previous section. Now there are four more saddle points, which are the corners of square on $(\omega_1, \omega_2)$ plane over which we are doing the $\omega_i$ integrals. The fluctuation around this each corner provides 1/4 of the contribution coming from the fluctuation around saddle $(0,0)$. This is self evident because a full circle around $(0,0)$

contributes to fluctuation integral whereas, only a quarter chunk of the circle contributes for the corner points. The leading value of the integral from all of the saddles is the same. So we have following expression for $H(\beta)$

$$
\begin{aligned}
H(\beta) &= \sum_{\text{saddles}} PE(\beta, \omega_i = \text{saddle value}) \times \text{fluctuation contribution} \\
&= \exp\left[\frac{\pi^4}{45}\beta^{-3} + \zeta'(-2)\right](1 + 4 \times 1/4) \\
&\qquad \left(\int_{-\infty}^{\infty} d\omega_1 \mu(\omega_1)\int_{-\infty}^{\infty} d\omega_2 \mu(\omega_2)\exp\left[-\frac{8\pi^6}{45\beta^5}(\omega_1^2 + \omega_2^2)\right]\right) \\
&= \frac{91125\beta^{15}}{16\pi^{13}}\exp\left[\frac{\pi^4}{45}\beta^{-3} + \zeta'(-2)\right].
\end{aligned}
\tag{124}
$$

When we add fermions in the mix, the saddle $\alpha = \gamma = -1$ produces a suppressed contribution at leading order (these are depicted by the dashed green lines in fig. 3). This is evident from the character of fermion being $(\alpha + \alpha^{-1}) - q(\gamma + \gamma^{-1})$. Thus we recover the result presented in Eq. (116).

**Take II—The $SO(4)$ Way**

$H(\beta)$ is given by

$$
H(\beta) = \oint dx_1 \oint dx_2 \mu_{SO(4)}(x_i)\exp\left[\sum_{n=1}^{\infty}\frac{e^{-n\beta}}{n}\frac{1 - e^{-2n\beta}}{P(n\beta, x_i^n)}\right],
\tag{125}
$$

where

$$
\mu_{SO(4)}(x_i) = \frac{1}{x_1 x_2}(1 - x_1 x_2)(1 - x_1/x_2),
\tag{126}
$$

and

$$
P(\beta, x_i) = \left(1 - e^{-\beta}x_1\right)\left(1 - e^{-\beta}x_1^{-1}\right)\left(1 - e^{-\beta}x_2\right)\left(1 - e^{-\beta}x_2^{-1}\right).
\tag{127}
$$

Now note that in the $\beta \to 0$ limit, the singularity appears when $x_1 = x_2 = 1$. In terms of the angular variable $\tilde{\omega}_i$ (where $x_i = e^{2\pi\iota\tilde{\omega}_i}$) we have the following saddle (the only one)

$$
(\tilde{\omega}_1, \tilde{\omega}_2) = (0, 0)
\tag{128}
$$

$$
\begin{aligned}
H(\beta) &= PE(\beta, \tilde{\omega}_i = \text{saddle value}) \times \text{fluctuation contribution} \\
&= \exp\left[\frac{\pi^4}{45}\beta^{-3} + \zeta'(-2)\right]\left(\int_{-\infty}^{\infty} d\tilde{\omega}_1 \int_{-\infty}^{\infty} d\tilde{\omega}_2 \mu_{SO(4)}(\omega_i)\exp\left[-\frac{4\pi^6}{45\beta^5}(\omega_1^2 + \omega_2^2)\right]\right) \\
&= \frac{91125\beta^{15}}{16\pi^{13}}\exp\left[\frac{\pi^4}{45}\beta^{-3} + \zeta'(-2)\right].
\end{aligned}
\tag{129}
$$

Thus Eq. (124) matches with Eq. (129). Note that we can not add fermions in the $SO(4)$ language—first we need to go to the double cover, see App. C for the details.

## 6.2 A phase transition in Hilbert series

In this section, we take a closer look at the $\beta \to 0$ behaviour of Hilbert series as a function of $\dim_f$. In particular, we want to inspect how the following comes about

$$H(\beta \to 0, \dim_f = 0) = 2\left( \lim_{\dim_f \to 0} \left[ H(\beta \to 0, \dim_f) \right] \right). \tag{130}$$

Let us consider the Hilbert series for $b$ scalars and $f$ spin-1/2 fermions,

$$H(\beta, f) = \oint d\alpha \oint d\gamma \left[ \frac{1}{\alpha\gamma}(1 - \alpha^2)(1 - \gamma^2) \right]$$
$$\times \exp\left[ \sum_{n=1}^{\infty} \frac{1}{n} \frac{be^{-n\beta}(1 - e^{-2n\beta}) + (-1)^{n+1}f\, e^{-3n\beta/2}\left[ (\alpha^n + \alpha^{-n}) - e^{-n\beta}(\gamma^n + \gamma^{-n}) \right]}{P(e^{-n\beta}, \alpha^n, \gamma^n)} \right]. \tag{131}$$

In the $\beta \to 0$ limit, the singularity appears when $\alpha = \gamma = \pm 1$. When $f \neq 0$, the leading singularity is $\alpha = \gamma = 1$, and the subleading one is $\alpha = \gamma = -1$. Let us keep them both. Now we have (we denote the Hilbert series as $H'$ to distinguish it from the one where we take only the leading saddle)

$$H'(\beta \to 0, f) = \frac{91125\beta^{15}}{32\pi^{13}(b + 7/8f)^3} \exp\left[ \frac{\pi^4(b + 7/8f)}{45\beta^3} - \frac{f\,\pi^2}{48\beta} + b\zeta'(-2) \right]$$
$$+ \frac{91125\beta^{15}}{32\pi^{13}(b - f)^3} \exp\left[ \frac{\pi^4(b - f)}{45\beta^3} + \frac{\pi^2 f}{24\beta} + b\zeta'(-2) \right]. \tag{132}$$

Now one can see that for $f = 0$ the two saddles coincide and instead of Eq. (130) we have

$$H'(\beta \to 0, f \to 0) = H'(\beta \to 0, f = 0). \tag{133}$$

So the apparent discontinuity/phase transition in Eq. (130) gets resolved by adding contribution from the second saddle; the second saddle becoming equally important when $f = 0$ We also remark that for $b \leq f$, the other saddle is unstable one, and one should not include it.

## 6.3 Saddles of internal groups

When taking projecting to singlets of some internal symmetry group, similar saddle point subtleties can arise. We already mentioned one in the case of a $U(1)$ symmetry in footnote 6. Another example: consider the Hilbert series for a single field transforming in the adjoint representation of $SU(N)$. In this case there are $N$ saddles that contribute equally. However, the inclusion of another field transforming in e.g. the fundamental representation suppresses all but one saddle (at the centre of the hypercube swept out by the angular fugacities), and a similar discontinuity to the one above occurs. We leave a detailed study of such discontinuities in Hilbert series to future work.

# 7 The fate of the SMEFT

We finally turn to applying our results to a a complicated phenomenological EFT, namely the SMEFT. This has field content shown in Table 1 (conjugates of all fields must also be included). We will consider this theory with $N_g$ copies of fermionic generations (the SM has $N_g = 3$). Hilbert series methodology was applied to this theory in [3] to systematically enumerate operators at mass dimension eight and above (results up to mass dimension 15 were presented).

Let us assemble the components that form the leading behaviour of the PE for the SMEFT in the $\beta \to 0$ limit.

Table 1: The field content of the SMEFT wiith representations under the Lorentz group $SU(2)_L \times SU(2)_R$, and the gauge group $SU(3)_c \times SU(2)_W \times U(1)_Y$ of the SM.

| Field | $SU(2)_L$ | $SU(2)_R$ | $SU(3)_c$ | $SU(2)_W$ | $U(1)_Y$ |
|---|---|---|---|---|---|
| $Q$ | **2** | **1** | **3** | **2** | 1/6 |
| $L$ | **2** | **1** | **1** | **2** | $-1/2$ |
| $u_c$ | **2** | **1** | $\overline{\mathbf{3}}$ | **1** | $-2/3$ |
| $d_c$ | **2** | **1** | $\overline{\mathbf{3}}$ | **1** | 1/3 |
| $e_c$ | **2** | **1** | **1** | **1** | 1 |
| $G_L$ | **3** | **1** | **8** | **1** | 0 |
| $W_L$ | **3** | **1** | **1** | **3** | 0 |
| $B_L$ | **3** | **1** | **1** | **1** | 0 |
| $H$ | **1** | **1** | **1** | **2** | 1/2 |

### Leading exponential

The leading exponential piece can be read straight from Eq. (116),

$$\log PE(\beta) \ni \left[ A\beta^{-3} + B\beta^{-1} \right] + C\zeta'(-2) + D\log(\beta/2\pi), \tag{134}$$

with

$$A = \left( \frac{\pi^4}{45}\dim_B + \frac{7\pi^4}{360}\dim_f \right), \; B = -\left( \frac{\pi^2}{48}\dim_{1/2} + \frac{\pi^2}{6}\dim_1 \right), \; C = \dim_B, \; D = -\frac{1}{2}\dim_1, \tag{135}$$

where from Table 1 (remembering to count the dof in the gauge group representations, and to include the conjugate fields) we have,

$$\dim_B = 28, \; \dim_f = 30N_g, \; \dim_1 = 24, \; \dim_{1/2} = 30N_g. \tag{136}$$

### Projection onto Lorentz scalars and IBP

These pieces can also be read straight from Eq. (116), and we find a suppression by a factor of

$$\frac{91125\beta^{15}}{32\pi^{13}} \left( \dim_B + \frac{7}{8}\dim_f \right)^{-3}, \tag{137}$$

for the projection to scalars, and factor of $\beta^4$ for from the IBP projector.

### Projection onto $U(1)$ singlets

We have the suppression found in Eq. (115), by a factor of

$$\sqrt{\frac{3}{2\pi^3}} \frac{\beta^{3/2}}{Q}, \quad \text{where } Q^2 \equiv \left( \sum_i q_{bi}^2 + \frac{1}{2}\sum_i q_{fi}^2 \right). \tag{138}$$

For the SMEFT, scaling the charges that appear in Table 1 by a factor of 6 (see footnote 6), we have

$$Q^2 = 36 + 120N_g. \tag{139}$$

**Projection onto $SU(2)$ singlets**

There is a the single saddle point at $\omega = 0$, and the integral over fluctuation is found to be

$$
\int_{-\infty}^{\infty} d\mu_{SU(2)}(\omega) \exp\left[-\frac{4\pi^4\omega^2}{3\beta^3}\left(\dim_B^{\text{fund}} + \frac{1}{2}\dim_f^{\text{fund}} + 4\dim_b^{\text{adj}}\right)\right]
$$
$$
= 3\sqrt{\frac{3}{4\pi^7}}\left(\dim_B^{\text{fund}} + \frac{1}{2}\dim_f^{\text{fund}} + 4\dim_B^{\text{adj}}\right)^{-3/2}\beta^{9/2}.
\tag{140}
$$

Here $\dim_B^{\text{fund}}$ counts the number of bosonic dof in the fundamental representation (rep) of $SU(2)$; $\dim_f^{\text{fund}}$ counts the number of fermionic dof in the fundamental rep of $SU(2)$; $\dim_B^{\text{adj}}$ counts the number of bosonic dof in the adjoint rep of $SU(2)$. For the SMEFT we have

$$
\dim_B^{\text{fund}} = 2, \quad \dim_f^{\text{fund}} = 8N_g, \quad \dim_B^{\text{adj}} = 2.
\tag{141}
$$

**Projection onto $SU(3)$ singlets**

There is a the single saddle point at $k_1 = k_2 = 0$, and the integral over fluctuation is found to be

$$
\int_{-\infty}^{\infty} d\mu_{SU(3)}(k_1, k_2) \exp\left[-\frac{4\pi^4(k_1^2 - k_1 k_2 + k_2^2)}{6\beta^3}\left(\dim_f^{\text{fund}} + \dim_{\bar{f}}^{\text{fund}} + 12\dim_b^{\text{adj}}\right)\right]
$$
$$
= \frac{432\sqrt{3}\beta^{12}}{\pi^9}\left(\dim_f^{\text{fund}} + \dim_{\bar{f}}^{\text{fund}} + 12\dim_B^{\text{adj}}\right)^{-4},
\tag{142}
$$

where $\dim_B^{\text{fund}}$ now counts the number of dof in the fundamental representation of $SU(3)$, etc.. For the SMEFT, we have

$$
\dim_f^{\text{fund}} = \dim_{\bar{f}}^{\text{fund}} = 4N_g, \quad \dim_B^{\text{adj}} = 2.
\tag{143}
$$

## 7.1 Putting it all together

Assembling the above components and the IBP suppression, we can construct the Hilbert series for the SMEFT in the $\beta \to 0$ limit:

$$
H(\beta) \underset{\beta \to 0}{\sim} \frac{44286750\beta^{25}\exp\left(\frac{7\pi^4}{180\beta^3}(15N_g + 16) - \frac{\pi^2}{8\beta}(5N_g + 32) + 28\zeta'(-2)\right)}{343\pi^{15}(N_g + 3)^4(2N_g + 5)^{3/2}\sqrt{10N_g + 3}(15N_g + 16)^3}.
\tag{144}
$$

Performing the inverse Laplace transform, one obtains the asymptotic growth of operators in the SMEFT

$$
\rho(\Delta) \underset{\Delta \to \infty}{\sim} \mathcal{N} \exp\left(\frac{2\pi\sqrt{2}}{3}\sqrt[4]{7N_g + \frac{112}{15}}\Delta^{3/4} - \frac{\pi(5N_g + 32)}{4\sqrt{2}\sqrt[4]{7N_g + \frac{112}{15}}}\Delta^{1/4} + 28\zeta'(-2)\right), \tag{145}
$$

where $\mathcal{N}$ is given by

$$
\mathcal{N} = \frac{27783\left(\frac{7}{5}\right)^{3/8}3^{5/8}\pi^{10}(15N_g + 16)^{27/8}}{1024000\sqrt[4]{2}\Delta^{55/8}(N_g + 3)^4(2N_g + 5)^{3/2}\sqrt{10N_g + 3}}.
\tag{146}
$$

We note that in performing the inverse Laplace transform, there is a simple way to probe the effect of higher order terms in the asymptotic series we present above. The transform is

$$
\int d\beta\, H(\beta)e^{\beta\Delta}, \tag{147}
$$

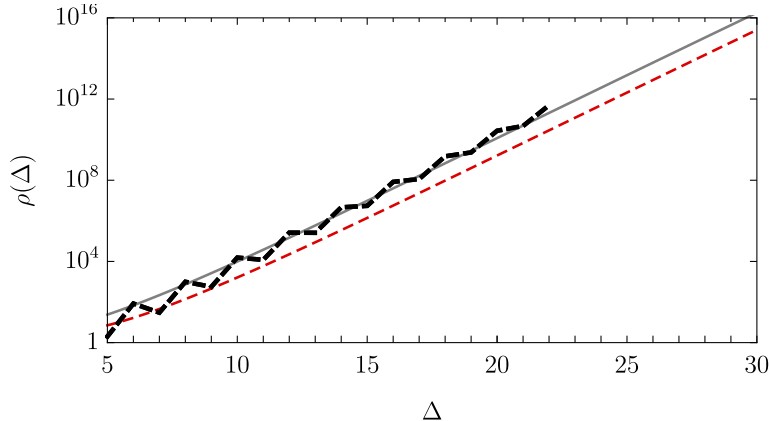

Figure 4: The effect of a linear shift in $\Delta$ to capture sub-leading corrections to the asymptotic formula for the growth of operators in the SMEFT, with one generation of fermions, $N_g = 1$. The thick dashed line is the exact results for the number of operators in the SMEFT as a function of $\Delta$. The thin dashed curve indicates the asymptotic result given in Eq. (145). The solid grey curve indicates Eq. (145) evaluated with the shifted $\widetilde{\Delta}$ given by Eq. (149).

where $H(\beta)$ contains the exponential term

$$H(\beta) \ni e^{A\beta^{-3} + B\beta^{-1} + C\beta^0 + D\log(\beta/2\pi) + \cdots}. \tag{148}$$

Here the $+\ldots$ denotes terms which vanish as $\beta \to 0$. The term linear in $\beta$, i.e. $+K\beta$ in the above exponent, generates an effective shift in $\Delta$ as can be seen from Eq. (147). That is, including the linear term one can define an effective $\rho(\widetilde{\Delta}) = \rho(\Delta + K)$ that captures its effect. Note that this implements sub-leading corrections to the asymptotic result, which will change the result at low, finite $\Delta$, but not as $\Delta \to \infty$.

When using the (non-rigorous) new trick of Sec. 3 we often kept such higher order terms in $\beta$ in the exponential at intermediate stages. By applying the trick to the SMEFT, and retaining terms up to linear order in $\beta$, we find an effective shift in $\Delta$

$$\widetilde{\Delta} = \Delta + \frac{67}{60} + \frac{17N_g}{64}. \tag{149}$$

We have observed experimentally that when the number of degrees of freedom in the EFT is large, such as it is for the SMEFT, this linear shift obtained by the trick brings the asymptotic result in better agreement with low scaling dimension data. (When the dof are small, it has little effect.) We do not speculate on why this is so here; our errors are not under control, precluding us from quantitative analysis. Nevertheless, it is interesting that this shift implements subleading corrections to Eq. (145) that bring it into remarkable agreement with low mass dimension data. This is illustrated in Fig. 4. The thick dashed line corresponds to the exact number of operators as a function of mass dimension $\Delta$ in the SMEFT, for the case of $N_g = 1$. In obtaining the exact results for the SMEFT beyond mass dimension 15 that are used in Fig. 4, as well as Fig. 5 below, we used the computer code accompanying the paper [53]. The thin dashed red curve indicates the result of Eq. (145), and the solid grey curve indicates Eq. (145) evaluated with the shifted $\widetilde{\Delta}$ given by Eq. (149).

Fig. 5 compares our results to exact data in the SMEFT for $N_g = 1$ and $N_g = 3$ (the full SMEFT), in both cases using Eq. (145) evaluated with the shifted $\widetilde{\Delta}$ in Eq. (149). Given that our formulae are only asymptotic, the agreement to exact data at low mass dimensions is striking. The agreement for $N_g = 3$ appears even better, in line with our observations that the

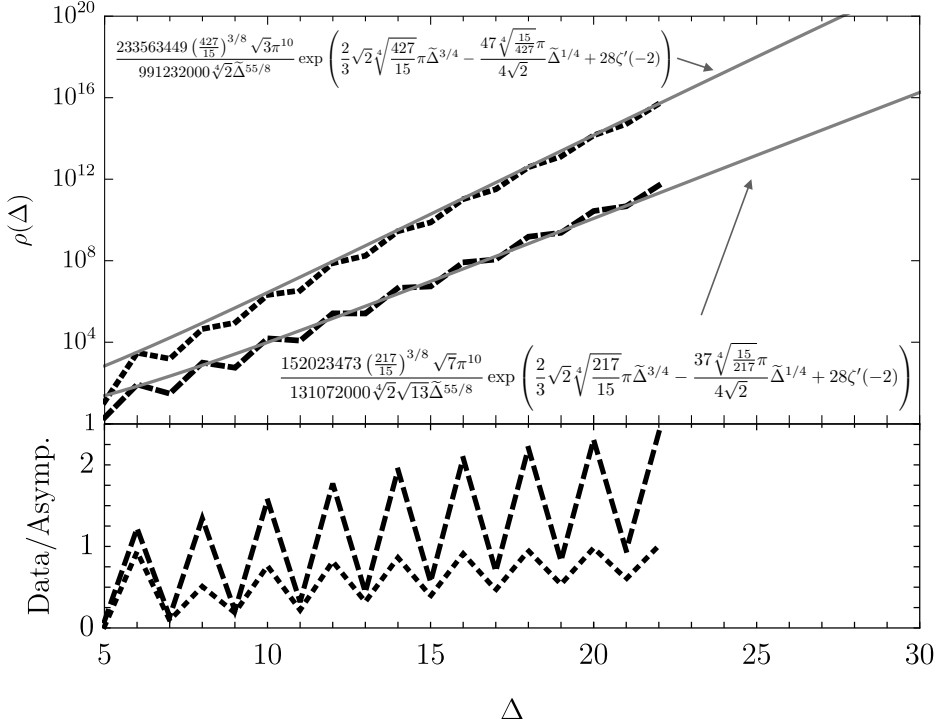

Figure 5: The growth of operators in the SMEFT, comparing exact data with the asymptotic formulae. The thick dashed curve is data for one generation of fermions, $N_g = 1$; the thick dotted curve is data for three generations of fermions $N_g = 3$. The solid grey curves indicate the asymptotic result Eq. (145) evaluated with the shifted $\widetilde{\Delta}$ given by Eq. (149).

shift works well for a large number of dof. However, such statements are only qualitative; we leave a quantitative assessment of error terms to future work.

## 8   Discussion

We have developed new techniques for studying EFTs/$S$-matrices, probing the asymptotic growth of operators through a study of the analytic behaviour of Hilbert series. Our methods also revealed phase transition-like discontinuities in evaluating projections onto singlets of Lorentz and internal symmetry groups. These results are a further step in the broader direction of using Hilbert series to study EFTs analytically. We end with a discussion of a few directions we can see to pursue so as to generalise and improve upon the above techniques.

Firstly, it would be interesting to study the asymptotics of finer-grained Hilbert series— the behaviour of the growth of operators projected onto different spin representations, or as a function of a fixed particular number of fields in an operator (see Sec 5 of [28]). The latter is particularly interesting from the point of view of studying $S$-matrix elements, as is corresponds to fixing the number external legs. If it is in any way possible to study of the convergence properties of the $S$-matrix of a dynamic theory by methods that mirror the known results about convergence of the OPE in CFT [54–57], our results on the growth of operator degeneracy must enter at some level.

The Hilbert series appears in disguise in the calculation of superconformal index [58] and partition functions in large $N$ quantum field theories [59, 60]. One often looks at large $N$ and/or high temperature behavior of the free energy of these theories, that can be obtained

from the partition function. From this vantage point, the Hilbert series is in fact closely related to the physical partition function on $S^1 \times S^{d-1}$ via radial quantization, where $S^1$ is the thermal circle and the $\beta \to 0$ behavior of the Hilbert series is nothing but the high temperature behavior of the partition function. Thus our result can be thought of as a study of growth of number of states in free CFT on a cylinder. In particular, we have performed a refined study of growth of all the states, the states sitting in various singlet sectors (singlet under the Lorentz group and/or internal symmetry group) of the theory.

Results on the leading term in the high temperature partition function for free CFTs (our Sec. 4.3) have appeared many times in the literature as a function of the number of bosonic or grassmannian variables of the theory under consideration. This began with the work of Cardy [32], and more recently with explorations in a holographic context in [61], and was again revisited in [62–64]. These papers find the same leading scaling of $\log H(\beta)$ in the $\beta \to 0$ limit as us. Our contribution is to pinpoint the the order one coefficients accompanying the leading term in $\log H(\beta)$ as a function of spin and statistics of the field content in any dimension. Furthermore, the refined analysis where we project onto the singlet sectors of various symmetry group required us to evaluate the subleading corrections with precise coefficients as well. To add to list of what have been explored in the literature, we note the application of Meinardus theorem in context of counting of BPS operators in [65], the apperance of Cardy formula in supersymmetric theories in [66] and a recent revival of Cardy like formulas in [67–70]. It would be nice to investigate whether some of the results appearing here would be useful in that context.

Our results can be quite generally applied to partition functions; for instance it would be interesting to apply them to Hilbert series of non-relativistic EFTs [29], and applications to theories of e.g. superfluidity [71] and more general condensed matter systems [72], which might exhibit a different high temperature behaviour. It would also be interesting to work out Meinardus theorems when other fugacities are present in the partition.

How can the accuracy and robustness of our formulas be improved? This question is particularly motivated given that *(i)* we already see remarkably encouraging agreement even at low mass dimension e.g. in the SMEFT such that one might wonder how good they can get, and *(ii)* results in the mathematics literature improve upon the Meinardus formula for the plane partitions $PL(n)$ considered in Sec. 3 to obtain integer results within error up to partitions on the order of $n \lesssim 6400$, (a number of around 300 digits) [73]. (Such precision would even be useful to turn the asymptotic formulas into a calculational tool in lieu of a Taylor series expansion of a Hilbert series to obtain the exact projection onto invariants.)

To elaborate further, we note that the actual density of states/operators is a distribution that has support on a set of discrete values. It is more appropriate to smear this distribution over a small window and estimate the number of operators with dimension lying in that window by establishing a rigorous upper and lower bound. In 2D CFT, this has been achieved in a series of papers [38–41], with a result of the following form,

$$(2\delta - 1)\rho_0(\Delta) \leq \int_{\Delta-\delta}^{\Delta+\delta} d\Delta' \rho(\Delta') \leq (2\delta + 1)\rho_0(\Delta), \tag{150}$$

where we have smeared the density of operators $\rho(\Delta')$ over a window centred at $\Delta$ with width $\delta$ and taken the limit $\Delta \to \infty$ with $\delta$ fixed. Here $\rho_0(\Delta)$ is the smooth approximation of $\rho(\Delta)$ which can be naively obtained by doing inverse Laplace transformation; the above equation immediately tells us that it is accurate up to order one multiplicative error.

One might wonder whether one needs to smear over a window for the asymptotic analysis that we have done, thus losinig control over the order one number. It is helpful to get some intuition from scenarios in 2D where such smearing is not actually needed. Famous examples of this include the growth of bosonic excitaions in 2D free boson CFT (which is related to the

integer partitions [44] and the 2D Hilbert series in Sec. 2), and the growth of operators in extremal CFTs (holomorphic CFTs with $c = 24k$ where $k$ is a positive integer; for $k = 1$ the partition function is given by $j$ function [74]). In such cases, one can obtain a convergent Rademacher sum [75,76] for the exact degeneracy of operators. The technical reason behind such an improvement over a generic CFT is the fact that we know the operators are regularly spaced (gapped by an integer). (This goes in other direction as well. In generic 2D CFTs, one can show the Cardy inequality is saturated iff we have integer spaced spectra asymptotically, which in turn implies all the operators are gathered together at some integer spaced points with huge degeneracy [41].)

The examples in higher dimension that we consider here are of this kind: the operator spectrum is regularly gapped. Thus one can hope to write down a formula without smearing and hope to get the order one number correct. The only way to rigorously estimate this is to figure out correction terms and show that they are suppressed e.g. by following the proof of Meinardus theorem (see chapter 6 in [50]) keeping track of the error terms. We remark that we have adapted and generalized only a part of the Meinardas theorem for our purposes, omitting the part where one estimates the error rigorously in a step analogous to doing inverse Laplace transformation. A more ambitious goal would be to write down something like a convergent Rademacher sum in higher dimension. We have not attempted either of the above approaches in this paper, but we expect that one can improve upon our result.

## Acknowledgements

T.M. and S.P. thank UCSD for support and hospitality where this work was initiated. We thank Aneesh Manohar and Masahito Yamazaki for comments on the manuscript. T.M. is grateful to Brian Henning, Xiaochuan Lu, and Hitoshi Murayama for many important and fruitful discussions about Hilbert series. In obtaining the exact results for the SMEFT beyond mass dimension 15, we acknowledge use of the form computer code accompanying the paper [53]. T.M. is supported by the World Premier International Research Center Initiative (WPI) MEXT, Japan, and by JSPS KAKENHI grants JP18K13533, JP19H05810, JP20H01896 and JP20H00153. S.P. acknowledges the support from Ambrose Monell Foundation and DOE grant DE-SC0009988.

## A   The effect of equations of motion on asymptotics

We illustrate the effect of EOM on the asymptotic growth of operators with the example of a single real scalar field in $d = 4$. Without EOM imposed, the PE is

$$PE_{\text{No EOM}} = \exp\left[\sum_{n=1}^{\infty} \frac{e^{-\beta n}}{n} \frac{1}{(1-e^{-\beta n})^4}\right]. \tag{151}$$

Applying the trick introduced in Sec. 3, keeping only singular terms in $\beta$, we find

$$\lim_{\beta \to 0} PE_{\text{No EOM}}(\beta) = \exp\left[\frac{\zeta(5)}{\beta^4} + \frac{\pi^4}{90\beta^3} + \frac{\zeta(3)}{3\beta^2} + \frac{19\log(\beta)}{720} - \frac{43}{288}\right]. \tag{152}$$

Next consider a more general partition

$$PE(K,\beta) = \exp\left[\sum_{n=1}^{\infty} \frac{e^{-\beta n}}{n} \frac{1-e^{-K\beta n}}{(1-e^{-\beta n})^4}\right], \tag{153}$$

with the case $K = 2$ corresponding to the PE for a real scalar field in $d = 4$ with EOM imposed. Again keeping only singular terms in $\beta$ in applying the trick of Sec. 3, the leading behaviour as $\beta \to 0$ is,

$$
\begin{aligned}
\lim_{\beta \to 0} PE(K, \beta) = \exp\Bigg[ &\frac{\pi^4 K}{90\beta^3} - \frac{1}{2\beta^2}(K-2)K\zeta(3) + \frac{1}{36\beta}\pi^2(K-2)(K-1)K \\
&+ \frac{1}{24}(K-2)^2 K^2 \log(\beta) + \frac{1}{72}\left(-12K^3 + 45K^2 - 46K\right) \Bigg].
\end{aligned}
\tag{154}
$$

The behaviour of this general partition function is exponentially suppressed for any finite value of $K$ compared to the PE for the scalar field without EOM imposed. Note that all sub-leading terms with $\beta$ dependence vanish when setting $K = 2$ to recover the physical case of imposing EOM

$$
\lim_{\beta \to 0} PE(K = 2, \beta) = \exp\left[ \frac{\pi^4}{45\beta^3} - \frac{1}{9} \right].
\tag{155}
$$

# B   Meinardus theorem in arbitrary $d$, $j$

In this appendix we collect results not presented in the main text that are obtained by generalizing Meinardus' theorem and applying it to EFTs in spacetime dimensions $d$ and with fields of spin $j$.

## B.1   Scalar field theory in $d = 2n + 1$ for $n \geq 1$

In $d = 2n + 1$ dimension, the canonical dimension of the scalar field is half integer. Thus we can not directly apply Meinardus theorem (or Lemma 6.1 or its modified form Eq. (80)). A different kind of modification is needed. This is somewhat analogous to the case of fermions in even dimensions.

We recast the PE in following way:

$$
\begin{aligned}
PE(\beta) = \prod_{n=0}^{\infty} \left( 1 - q^{n + \frac{d-2}{2}} \right)^{-f(n,d)} &= \prod_{k=1}^{\infty} \left( 1 - q^{k-1/2} \right)^{-f(k-(d-1)/2, d)} \\
&= \frac{\prod_{k=1}^{\infty} \left( 1 - (\sqrt{q})^k \right)^{-\tilde{f}(k,d)}}{\prod_{k=1}^{\infty} \left( 1 - (\sqrt{q})^{2k} \right)^{-\tilde{f}(2k,d)}},
\end{aligned}
\tag{156}
$$

where $f(n, d)$ is given by the symmetric spin $n$ respresentation of $SO(d)$ i.e.

$$
f(n, d) = \dim[n, 0, 0, \cdots].
\tag{157}
$$

In the second step, we performed a change of variable and defined $\tilde{f}(k, d) = f(k/2 + 1/2 - \frac{d-1}{2}, d)$. This is constructed in a way such that $\tilde{f}(2k - 1, d) = f(k - (d-1)/2, d)$. Note, in the second equality, shift in $k$ is valid since $f(k - (d-1)/2, d) = 0$ for $1 \leq k < (d-1)/2$ and $k$ being integer. Thus the PE can be written as a ratio of two auxiliary PE, on which one can apply the modified Meinardus theorem:

$$
PE(\beta) = \frac{PE^{aux1}(\beta/2)}{PE^{aux2}(\beta)},
\tag{158}
$$

$$PE^{aux1}(\beta) = \prod_{k=1}^{\infty} \left(1 - e^{-k\beta}\right)^{-f\left(k/2 + 1/2 - \frac{d-1}{2}, d\right)},$$

$$PE^{aux2}(\beta) = \prod_{k=1}^{\infty} \left(1 - e^{-k\beta}\right)^{-f\left(k + 1/2 - \frac{d-1}{2}, d\right)}.$$

(159)

The $D$ functions corresponding to the auxiliary PEs are found to be

$$D_1(d,s) = \sum_k k^{-s} f\left(k/2 + 1/2 - \frac{d-1}{2}, d\right),$$

$$D_2(d,s) = \sum_k k^{-s} f\left(k + 1/2 - \frac{d-1}{2}, d\right),$$

(160)

and let us denote the poles as $\alpha_{i,1}, \alpha_{j,2}$ with residues $A_{i,1}$ and $A_{j,2}$ where the index 1 and 2 refer to the $PE^{aux1}$ and $PE^{aux2}$. In terms of these variables we have

$$\log PE(\beta \to 0) = \log PE^{aux1}(\beta/2 \to 0) - \log PE^{aux2}(\beta \to 0)$$

$$= \left(\sum_i A_{i,1} \Gamma(\alpha_{i,1}) \zeta(\alpha_{i,1} + 1)(\beta/2)^{-\alpha_{i,1}}\right) - \left(\sum_j A_{j,2} \Gamma(\alpha_{j,2}) \zeta(\alpha_{j,2} + 1)\beta^{-\alpha_{j,2}}\right)$$

$$- \left[D_1(d,0) \log(\beta/2) - D_2(d,0) \log(\beta)\right] + D_1'(d,0) - D_2'(d,0).$$

(161)

**d=3**

Let us do the case $d = 3$ explicitly (we have done this before using Wright's result for plane partitions in Sec. 3; here we employ a method generalizable to arbitrary odd dimension). We have the following data

$$D_1(3,s) = \zeta(s-1), \quad D_2(3,s) = 2\zeta(s-1),$$
$$\alpha_{1,1}(3) = 2, \quad \alpha_{2,1}(3) = 2$$
$$A_{1,1}(3) = 1, \quad A_{2,1}(3) = 2,$$

(162)

leading to Eq. (60).

**d=5**

For $d = 5$ we have

$$D_1(5,s) = \frac{1}{24}\left[\zeta(s-3) - \zeta(s-1)\right]$$
$$\alpha_{1,1}(3) = 4, \quad \alpha_{2,1}(3) = 2$$
$$A_{1,1}(3) = 1/24, \quad A_{2,1}(3) = -1/24,$$

(163)

and

$$D_2(5,s) = \frac{1}{12}\left[4\zeta(s-3) - \zeta(s-1)\right],$$
$$\alpha_{1,2}(5) = 4, \quad \alpha_{2,2}(5) = 2$$
$$A_{1,2}(5) = 1/3, \quad A_{2,2}(3,s) = -1/12.$$

(164)

Applying Eq. (161) we find

$$\log PE_{d=5}(\beta \to 0) = \frac{2\zeta(5)}{\beta^4} - \frac{\zeta(3)}{12\beta^2} + \frac{17\log(\beta)}{2880} + \frac{11}{2880}\log(2) - \frac{7}{24}\zeta'(-3) + \frac{1}{24}\zeta'(-1).$$

(165)

Using the trick, one can find that

$$\log PE_{d=5}^{\text{trick}}(\beta \to 0) = \frac{2\zeta(5)}{\beta^4} - \frac{\zeta(3)}{12\beta^2} + \frac{17\log(\beta)}{2880} + \frac{68\log\left(\frac{3}{2}\right) - 131}{11520}. \tag{166}$$

Now we have

$$\underset{\exp[(165)]}{\text{actual asymptotics}} : \underset{\exp[(166)]}{\text{asymptotics via new trick}} = 1 : 0.996. \tag{167}$$

## B.2 Fermionic field theory in $d = 2n+1$ for $n \geq 1$

The canonical dimension of the fermionic field is an integer. This mimics the case of scalars in even dimension. Thus we have

$$PE(\beta) = \prod_{n=0}^{\infty} \left(1 + q^{n+\frac{d-1}{2}}\right)^{g(n,d)} = \prod_{k=1}^{\infty} \left(1 + q^k\right)^{g(k-(d-1)/2,d)} \tag{168}$$

$$g(n,d) = \dim[n+1/2, 1/2, \cdots], \tag{169}$$

for example

$$g(n,3) = 2(n+1), \quad g(n,5) = \frac{2}{3}(n+1)(n+2)(n+3). $$

The limit in $k$ can be shifted to $k = 1$ in the above because $g(k-(d-1)/2,d) = 0$ for $k < (d-1)/2$ and $k \in \mathbb{Z}_+$.

Let us apply this on spin $1/2$ fermions in $d = 3$ and $d = 5$. Explicitly we have

$$\begin{aligned} D(3,s) &= 2\zeta(s-1), \quad \alpha(3) = 2, A(3) = 2 \\ D(5,s) &= \frac{2}{3}[\zeta(s-3) - \zeta(s-1)], \quad \alpha_1(5) = 4, \alpha_2(5) = 2, A_1(5) = -A_2(5) = \frac{2}{3}. \end{aligned} \tag{170}$$

Using the results following from Eq. (91), we obtain

$$\begin{aligned} \log PE_{d=3}^{(f)}(\beta) &\underset{\beta \to 0}{\simeq} \frac{3}{2}\zeta(3)\beta^{-2} - \frac{\log 2}{6} \\ \log PE_{d=5}^{(f)}(\beta) &\underset{\beta \to 0}{\simeq} \frac{15\zeta(5)}{4\beta^4} - \frac{\zeta(3)}{2\beta^2} + \frac{11}{180}\log(2). \end{aligned} \tag{171}$$

Here we have put in the supersprcipt $f$ explicitly to denote that these are fermionic PE. It can be verified that our trick reproduces these asymptotics exactly.

## B.3 Asymptotics of spin $j$ fields in $d = 4$

We focus on $d = 4$ dimensional field theories with arbitraty spin $j$ field, saturating the unitarity bound. Fields of spin $j$ have dimension $j+1$. So all the bosonic fields have integer dimension while the fermionic field have half-integer dimension.

**Bosonic fields** $j \in \mathbb{Z}$

The PE is given by

$$
\begin{aligned}
PE(j,\beta) &= \exp\left[\sum_{n=1}^{\infty} \frac{e^{-(j+1)n\beta}}{n}\left(\frac{2j+1-4je^{-n\beta}+(2j-1)e^{-2n\beta}}{(1-e^{-n\beta})^4}\right)\right] \\
&= \prod_{n=0}^{\infty}\left(1-e^{-(n+j+1)\beta}\right)^{-(n+1)(n+2j+1)} \\
&= \frac{\prod_{k=1}^{\infty}\left(1-e^{-k\beta}\right)^{-(k^2-j^2)}}{\prod_{k=1}^{j}\left(1-e^{-k\beta}\right)^{-(k^2-j^2)}},
\end{aligned}
\tag{172}
$$

where the factor $(n+1)(n+2j+1)$ comes from the dimension of the $SO(4)$ representation $[n+j,j]$.

The quantity in the denominator will produce a polynomial factor in $\beta$ in $\beta \to 0$ limit:

$$
\prod_{k=1}^{j}\left(1-e^{-k\beta}\right)^{-(k^2-j^2)} \underset{\beta\to 0}{\simeq} \prod_{k=1}^{j}(k\beta)^{j^2-k^2} = \beta^{\frac{1}{6}(j-1)j(4j+1)}\prod_{k=1}^{j}(k)^{j^2-k^2}.
$$

The numerator can be handled using Meinardus approach. The relevant function $D$ is given by

$$
D(j,s) = \sum_{k=1}^{\infty} k^{-s}(k^2-j^2) = \zeta(s-2)-j^2\zeta(s).
\tag{173}
$$

Hence we have

$$
\begin{aligned}
\log PE(j,\beta) \underset{\beta\to 0}{\simeq}\ & \Gamma(3)\zeta(4)\beta^{-3} - j^2\Gamma(1)\zeta(2)\beta^{-1} - \frac{j^2}{2}\log\beta + \zeta'(-2) + \frac{1}{2}j^2\log(2\pi) \\
& - \frac{1}{6}(j-1)j(4j+1)\log\beta - \sum_{k=1}^{j}(j^2-k^2)\log k \\
=\ & \frac{\pi^4}{45\beta^3} - \frac{\pi^2 j^2}{6\beta} - \frac{1}{6}j(2j-1)(2j+1)\log(\beta) + \zeta'(-2) + \frac{1}{2}j^2\log(2\pi) \\
& - \sum_{k=1}^{j}(j^2-k^2)\log k.
\end{aligned}
\tag{174}
$$

**Fermionic fields** $2j \in \mathbb{Z}/2\mathbb{Z}$

The PE is given by

$$
\begin{aligned}
PE(j,\beta) &= \exp\left[\sum_{n=1}^{\infty}(-1)^{n+1}\frac{e^{-(j+1)n\beta}}{n}\left(\frac{2j+1-4je^{-n\beta}+(2j-1)e^{-2n\beta}}{(1-e^{-n\beta})^4}\right)\right] \\
&= \prod_{n=0}^{\infty}\left(1+e^{-(n+j+1)\beta}\right)^{(n+1)(n+2j+1)} \\
&= \frac{\prod_{k=1}^{\infty}\left(1+e^{-(k-1/2)\beta}\right)^{(k-1/2)^2-j^2}}{\prod_{k=1}^{j+1/2}\left(1+e^{-(k-1/2)\beta}\right)^{(k-1/2)^2-j^2}} \\
&= \prod_{k=1}^{j+1/2}\left(1+e^{-(k-1/2)\beta}\right)^{j^2-(k-1/2)^2}\frac{\prod_{k=1}^{\infty}\left(1+e^{-k\beta/2}\right)^{k^2/4-j^2}}{\prod_{k=1}^{\infty}\left(1+e^{-k\beta}\right)^{k^2-j^2}}.
\end{aligned}
\tag{175}
$$

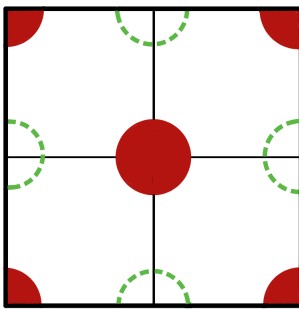

Figure 6: Calculation using the double cover of $SO(4)$. The figure depicts saddles on $(\tilde{\omega}_1, \tilde{\omega}_2)$ plane. The center of the square is at $(0,0)$. The corners are at $(1,1),(1,-1),(-1,1),(-1,1)$. For the bosonic case, all the saddles contribute. When we add fermions in the mix, the green coloured saddles at $(0,\pm1),(\pm1,0)$ don't contribute anymore, leading to an overall factor of $1/2$. We have drawn circular regions arounds the saddle to denote how much the fluctuation around the saddle contributes. For example, each of the corner ones contributes one quarter of the center one.

Thus we have

$$\log PE(j,\beta) \underset{\beta\to0}{\simeq} \frac{1}{6}j(2j-1)(2j+1)\log 2 + \lim_{\beta\to0}\log\frac{PE_1^{aux}(\beta/2)}{PE_2^{aux}(\beta)}. \tag{176}$$

We apply Meinardus theorem on $PE_1^{aux}$ and $PE_2^{aux}$. Now we have

$$D_1(j,s) = \frac{1}{4}\left[\zeta(s-2)-4j^2\zeta(s)\right], \quad D_2(j,s) = \left[\zeta(s-2)-j^2\zeta(s)\right]. \tag{177}$$

Thus we have

$$\log PE(j,\beta) \underset{\beta\to0}{\simeq} \frac{7\pi^4}{360\beta^3} - \frac{\pi^2 j^2}{12\beta} + \frac{1}{6}j(2j-1)(2j+1)\log 2. \tag{178}$$

## C    Combinatorical geometry for $SO(d)$ saddle points in arbitrary dimension

We begin with presenting a third take on the calculation of the saddle points of a real scalar in Sec. 6—doing the calculation in the $SO(4)$ double cover. We then show how the phase transition behaviour discussed in Sec. 6 occurs in arbitrary spacetime dimension.

### C.1    Take III—The $SO(4)$ double cover way

In this case, the contours must traverse the unit circle in the complex plane twiice. $H(\beta)$ is given by

$$H(\beta) = \frac{1}{4}\oint_{\text{twice}} dx_1 \oint_{\text{twice}} dx_2 \, \mu_{SO(4)}(x_i)\exp\left[\sum_{n=1}^{\infty}\frac{e^{-n\beta}}{n}\frac{1-e^{-2n\beta}}{P(n\beta,x_i^n)}\right], \tag{179}$$

where the factor of $\frac{1}{4}$ in front of the measure normalizes it to unity. As $x_1, x_2$ traverses the circle twice, we have a square region swept out on $(\tilde{\omega}_1, \tilde{\omega}_2)$ plane with center at $(0,0)$ and veritces at $(\pm1,\pm1),(\pm1,\mp1))$, and the positions of the saddles are at

$$(\omega_1,\omega_2) = \{(0,0),(0,1),(0,-1),(1,0),(-1,0),(1,1),(1,-1),(-1,1),(-1,-1)\}. \tag{180}$$

Again we have to sum over the saddles along with the fluctuations around it. Compared to the contribution coming from the fluctuation around the middle saddle $(0,0)$, the corners ones i.e $(\pm 1, 1), (1, \pm 1)$ produces $1/4$ of the contribution, while each of the $(0, \pm 1), (\pm 1, 0)$ produces $1/2$, see Fig. 6. So we have $(1 + 4 \times 1/4 + 4 \times 1/2) = 4$ coming from all the saddles and this kills the $1/4$ normalization factor appearing in front of the measure as a result of going to the double cover. In this way it matches with Eq. (124) and Eq. (129).

Adding fermions kills off the saddle at $(0, \pm 1), (\pm 1, 0)$, so now we have the center ones and the corner ones giving a contribution of $1 + 4 \times 1/4 = 2$; we must include again the overall factor of $1/4$ in the defining integral. Thus we land up with an overall factor of $1/2$ and reproduce the result in Eq. (116).

## C.2  Bosonic theory

There is only one saddle at $\tilde{\omega}_i = 0$ for the following integral:

$$H(\beta) = \oint \prod_i dx_i \, \mu_{SO(d)}(x_i) \exp\left[\sum_{n=1}^{\infty} \frac{e^{-n\beta}}{n} \frac{1 - e^{-2n\beta}}{P(n\beta, x_i^n)}\right]. \tag{181}$$

If we use the double cover then we have following expression in even $d$ dimensions

$$H(\beta) = \frac{1}{2^{d/2}} \oint_{\text{twice}} \prod_i dx_i \, \mu_{SO(d)}(x_i) \exp\left[\sum_{n=1}^{\infty} \frac{e^{-n\beta}}{n} \frac{1 - e^{-2n\beta}}{P(n\beta, x_i^n)}\right]. \tag{182}$$

We are going to show the extra $2^{d/2}$ factor gets killed by presence of other saddles. Now the saddle points are distributed over a hypercube of length 2 centered at $(0, 0, \cdots)$, aligned to the axes. There are $3^{d/2}$ saddle points, they are given by

$$\text{Saddles} = \{(a_1, a_2, \cdots a_{d/2}) : a_i \in \{0, \pm 1\}\}. \tag{183}$$

For $d = 4$, we have $3^{4/2} = 9$ points, they comes in three kinds (see fig Fig. 6 and Fig. 7). One point is at the center, four of them are at the midpoints of edges, and four of them are at the corners. The leading contribution from these saddles are the same, but the sub-leading contribution coming from the fluctuation around these three kinds of saddles is different. Geometrically, one can map these three kind of saddles with existence of geometrical objects, which are embedded in the 2 dimensional square (the integration region). We have exactly three such kinds: zero, one or two dimensional objects. The zero dimensional objects are the 4 corners, the one dimensional objects are the 4 edges, and the two dimension object is the full square. Thus the saddles are mapped to 1 two dimensional object, 4 one dimensional objects and 4 zero dimensional objects. This pattern survives in higher dimension. There are $d/2$ different type of saddles, which can be mapped to $d/2$ different type of dimensional objects which can embedded.

The number of $k$ dimensional objects inside a $d/2$ dimensional object is given by the expression $2^{d/2-k} \text{Binomial}[d/2, k]$. This follows because the $k$ dimensional object can be found by setting $k$ entries of $\{a_1, a_2, \cdots a_{d/2}\}$ to 0 and setting rest of them to $\pm 1$. The $k$ entries then scan be chosen in $\text{Binomial}[d/2, k]$ ways and rest of them can be filled in $2^{d/2-k}$ ways, leading to the above expression. Also, note that the compared to the contribution coming from the fluctuation around the saddle at the center (which is mapped to the $d/2$ dimensional object) saddles corresponding to the $k$ dimensional object contribute a factor of $2^{k-d/2}$. This can be obtained if we think coordinate-wise, the coordinates set at 0 contribute completely to the fluctuation, while the ones at $\pm 1$ contributes $1/2$ of the full contribution, thus leading to the

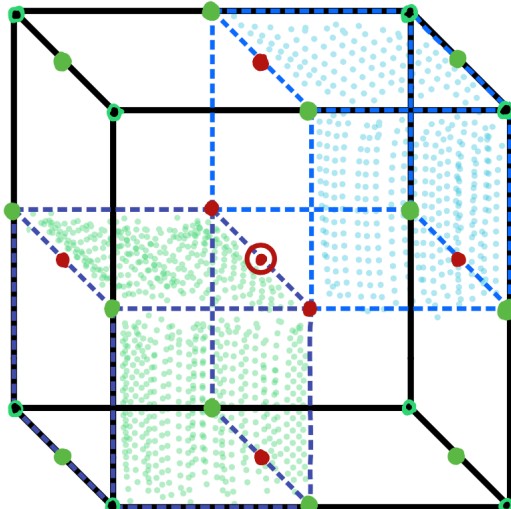

Figure 7: Calculation using double cover of $SO(6)$: the figure depicts saddles on $(\tilde{\omega}_1, \tilde{\omega}_2, \tilde{\omega}_3)$ 3−space. The center of the cube is at $(0,0)$, marked red and circled. The 8 corners are at $(a_1, a_2, a_3)$ where $a_i = \pm 1$. For the bosonic case, all the saddles contribute. When we add fermions in the mix, the green colored saddles at the middle point of edges (corresponding to 1 dimensional object edge) and green/black colored saddles at the corners (corresponding to 3 dimensional object cube) don't contribute anymore, leading to a factor of 1/2. The saddles corresponding to the zero dimensional center and midpoints of the faces (2 dimensional object) always contribute. Fluctuation around the saddles corresponding to $k$ dimensional object contributes $2^{k-3}$ compared to that of the center one.

$2^{k-d/2}$ suppression. We have a total contribution compared to the single cover

$$\frac{1}{2^{d/2}} \sum_{k=0}^{d/2} \underbrace{2^{d/2-k} \text{Binomial}[d/2, k]}_{\#\text{of saddles}} 2^{k-d/2} = 1 \,. \tag{184}$$

Now if we add fermions to the mix, some of the saddles will get suppressed at leading order compared to the one at the center. We will take up this problem in next subsection.

## C.3 Fermionic theory

For simplicity let us consider the spin $[1/2, \cdots 1/2]$ case in even dimension. The Hilbert series is given by

$$H(\beta) = \frac{1}{2^{d/2}} \oint_{\text{twice}} \prod_i dx_i \, \mu_{SO(d)}(x_i) PE(x_i, \beta)$$

$$PE \equiv \exp\left[\sum_{n=1}^{\infty} \frac{(-1)^{n+1} e^{-n\beta(d-1)/2}}{n} \frac{\chi_{[1/2, \cdots 1/2, +1/2]} - \chi_{[1/2, \cdots 1/2, -1/2]} e^{-n\beta}}{P(n\beta, x_i^n)}\right] \,. \tag{185}$$

The fermionic characters have branch-cuts. Thus the saddles that contribute to the leading order necessarily have the following form

$$\{a_1, a_2, \cdots a_{d/2} : a_i \in \{0, \pm 1\} \,\&\, \# \text{ of } \pm 1 \text{ entries is even}\} \,. \tag{186}$$

Thus compared to Eq. (184), we only sum over even $k$ i.e. we have total contribution

$$\frac{1}{2^{d/2}} \sum_{k=0}^{\lfloor d/4 \rfloor} \underbrace{2^{d/2-2k}\mathrm{Binomial}[d/2, 2k]}_{\#\text{of saddles}} 2^{2k-d/2} = \frac{1}{2}. \qquad (187)$$

Thus we arrive at

$$H(\beta \to 0, \dim_f = 0) = 2\left(\lim_{\dim_f \mapsto 0} \left[ H(\beta \to 0, \dim_f \right] \right). \qquad (188)$$

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
