# Peer review of "EFT Asymptotics: the Growth of Operator Degeneracy"

_SciPost Physics, doi:SciPost Phys. 10, 104 (2021)_

## Round 1 · Referee Report · Anonymous (Referee 1) · 2020-12-7

Strengths

1) Broad in terms of applications 2) Compact analytic formulae to compare with numerical results

Weaknesses

1) The manuscript is echnical and addressed principally to experts in the field 2) Applications a bit vague

Report

Dear Editor,

Using methods based on Hilbert series, the authors study the asymptotic growth of the number of operators in Effective Field Theories (EFTs) with different field content (with and without spin) and in different dimensions. The connection between the mathematical technique employed in the paper and the underlying physics is interesting. The result in terms of analytic formulae fills an important gap in a field that was (at his frontier) defined by numerical progress.

I feel that, moreover, the manuscript is addressing the more subtle question of whether Hilbert series are useful at all, beyond checking numerical calculations. The authors provide a number of interesting potential applications, though I feel these could have been given more weight and be used to shape the manuscript.

All in all the article meets the SciPost criteria in terms of interest. I nevertheless have a few questions and comments that I would like the authors to address :

1) The manuscript gives quite some importance to the notion of phase transitions (these are mentioned in the abstract, introduction and conclusions), while they occupy only a quarter of a page in the main text. What are the physics implications of this? In what space should it be considered a phase transition?

2) Can the authors be more precise (in the introduction) as to what concrete physics application these analytic results could have?

3) The authors introduce a non-rigorous trick in section 3 to compute the asymptotics, is this necessary? One of the motivations is that it is fast, but I could not understand why this would make such detour worth.

4) Mainardus theorem is never explicitly reported in the manuscript : is it really a theorem or simply a technique? In what sense it differes from Cardy’s approach?

Requested changes

In addition to the changes triggered by the comments in my report, I suggest the following small changes:

1) Below eq 1.1 mention that the PE is the generating functional

2) The number of partitions is introduced (very pedagogically) in section 3.2, while it is already used and discussed in section 2.2. Perhaps parts of section 3.2 could appear earlier.

3) I think ref 71 should be cited in the main text.

  • validity: top
  • significance: good
  • originality: high
  • clarity: good
  • formatting: excellent
  • grammar: perfect

Author:  Sridip Pal  on 2021-01-28  [id 1182]

(in reply to Report 1 on 2020-12-07)
Category:
remark
answer to question
correction

Dear Editor,

We thank the referee for their review of the manuscript and the comments and suggestions.

  1. It is a phase transition in the space of the (asymptotic growth of) degeneracy of operators. For example. one sees a non-continuous limit as the number of fermion species Nf, treated as a real parameter, goes to zero. Because it in the space of degeneracies (and also because the parameters, e.g. Nf here, are integers on physical grounds) we do not have a clear physics implication, and we have refrained from hinting at any such in the manuscript. However, we hope our use of ‘phase transition-like’ is useful language to visualize what is happening mathematically, that is, two maxima are becoming degenerate in the saddle approximation as Nf->0. This phase transition-like behavior is the subject of section 6 in the paper.

  2. We have added a paragraph to the introduction on other physics applications that these results could have.

  3. We believe that the trick it is useful (beyond just that of quick calculation—although we certainly value it that way ourselves) for being applicable to partition functions which are not immediately in the correct form for Meinardus’ theorem to be applied. In fact, the chronology in which we figured out the asymptotics went this way: we happened to come across Meinardus theorem after we used the trick. Having the correct inverse beta terms down to log(beta) served as a useful hint for massaging the PE into a form amenable to Meinardus. For these reasons, we hope its inclusion in the paper is valuable.

  4. Now we have modified the beginning of section 4 and beginning of subsection 4.1 clarifying the use of Meinardus theorem and its relation to Cardy’s paper.

All the minor suggestions are also taken care of. 1.We have added a line below Eq 1.1 2. We added a footnote in section 2.2 clarifying that the partition of integer discussed there is different from the plane partition discussion in section 3.2 3. We have moved the citation into section 7, when referring to the data points plotted in figs 4 and 5.

Best, Tom and Sridip

---

## Round 1 · Referee Report · Anonymous (Referee 2) · 2021-1-18

Report

I will not be able to accept this paper as it stands, with all the good will.
The authors start with a bad definition in equation 2.1 of a diverging quantity.
The PE is defined on a function which vanishes when all its variables are set to 0, and they fail to make sure of it.
Consequently, the PE of some function must start with 1, and again the authors fail to make sure of this.
There are many ways of working with finite expressions and there is a lot of literature which demonstrate how to do this.
The fact that the mistake happens at the very beginning makes it very hard to proceed, as it is based on a wrong definition of a basic function that is used throughout the paper.
From there the text goes downhill, as the authors are fixing this mistake with some “tricks” that are not acceptable.
One should remember that the PE is a combinatorial function that has no reason to diverge at any point of the computation.
In fact, as the authors are interested in a large order behavior, the origin is not so crucial, but this does not mean that they should start with a bad definition. One needs a minimal amount of modifications to keep the computations mathematically viable and with avoiding such sloppiness. After all this is not a path integral with infinitely many degrees of freedom.
Anyhow, if the authors want my approval, they will need to remove the ill definitions and to present their computations in a divergence less manner. Once this is fixed, I am happy to have another look at the paper.
  • validity: -
  • significance: -
  • originality: -
  • clarity: -
  • formatting: -
  • grammar: -

Author:  Sridip Pal  on 2021-01-28  [id 1181]

(in reply to Report 2 on 2021-01-18)
Category:
remark
answer to question
reply to objection
correction

Dear Editor,

We thank the referee for their review of the manuscript.

With regards to the point raised, indeed, for the scalar field in d=2, we started with an unregularized plethystic exponential in 2.1, but directly followed this with a paragraph describing the regularization procedure, along with its physical interpretation.

Our motivation for doing this was heuristic, and to try and walk through the regularization procedure from a physics point of view. However, we accept the referees point and have removed the unregularized expression, simply beginning with the regularized PE.

We stress that unregularized expressions only appeared in 2.1 and 2.4 of the paper, and only in the context of this heuristic discussion of the necessary regularization in d=2. That is, all parts of the paper from section 2.1 and beyond were (and remain) well-defined.

Best,
Tom and Sridip

---

## Round 2 · Referee Report · Anonymous (Referee 1) · 2021-2-15

Report

The authors addressed all pointsraised in my report, and I am ready to recommend the manuscript for SciPost publication.

---

## Editorial Decision

published